# scientific report

# Antisense regulation by transposon-derived RNAs in the hyperthermophilic archaeon *Sulfolobus solfataricus*

*Birgit Märtens[1], Salim Manoharadas[1], David Hasenöhrl[1], Andrea Manica[2] & Udo Bläsi[1+]*

[1]Department of Microbiology, Immunobiology and Genetics, Max F. Perutz Laboratories, and [2]Department of Genetics in Ecology, University of Vienna, Vienna, Austria

We report the first example of antisense RNA regulation in a hyperthermophilic archaeon. In *Sulfolobus solfataricus*, the transposon-derived paralogous RNAs, RNA-257$_{1-4}$, show extended complementarity to the 3′ UTR of the *1183* mRNA, encoding a putative phosphate transporter. Phosphate limitation results in decreased RNA-257$_1$ and increased *1183* mRNA levels. Correspondingly, the *1183* mRNA is faster degraded *in vitro* upon duplex formation with RNA-257$_1$. Insertion of the *1183* 3′ UTR downstream of the *lacS* gene results in strongly reduced *lacS* mRNA levels in transformed cells, indicating that antisense regulation can function in *trans*.
Keywords: *Sulfolobus solfataricus*; non-coding RNA; antisense regulation

## INTRODUCTION

In prokaryotes, small non-coding RNAs are involved in various biological processes, including transcriptional and translational regulation, RNA processing, RNA-guided modification of RNA and chromosome replication [1]. In eukaryotes, short interfering RNAs and microRNAs (miRNAs) act as regulators of development, cell death and chromosome silencing. Short interfering RNAs are derived from double-stranded RNA and act by RNA interference (RNAi) resulting in cleavage of the target mRNA [2]. The genome-encoded miRNAs act as components of ribonucleoprotein complexes. Binding of these complexes to the 3′ untranslated region (UTR) of mRNAs leads to translational repression and/or mRNA decay [3]. In contrast to eukaryal miRNAs, in bacteria, small regulatory RNAs (sRNAs) predominantly target the 5′ UTR of mRNAs [1].

[1]Department of Microbiology, Immunobiology and Genetics, Max F. Perutz Laboratories, University of Vienna, Dr Bohrgasse 9, 1030 Vienna, Austria
[2]Department of Genetics in Ecology, University of Vienna, Althanstrasse 14, 1090 Vienna, Austria
[+]Corresponding author. Tel: +43 1 427754609; Fax: +43 1 42779546;
E-mail: udo.blaesi@univie.ac.at

In bacteria, two major classes of RNAs involved in gene regulation can be discerned, *cis*- and *trans*-acting RNAs. The prototypic *trans*-acting sRNAs of Enterobacteriaceae have a typical size between 50 and 200 nucleotides, are usually not genetically linked to the loci of their target genes and are often expressed under specific growth or stress conditions [1,4]. Whereas some sRNAs act to modulate the activity of proteins, the majority appears to modulate gene expression by non-contiguous base-pairing with the 5′ UTR of mRNAs [1]. Regulation is mainly negative and seems to occur largely at the level of translation initiation and mRNA stability control [5]. *Cis*-acting bacterial RNAs either arise from short convergent transcripts that are complementary to the 5′ UTR and the immediate coding region of their target mRNA or from mRNAs containing a long 5′ or 3′ UTR that overlap with the mRNA encoded by the complementary DNA strand [5,6]. In bacteria, *cis*-antisense RNAs are involved in DNA replication control, maintenance of plasmids and in virulence gene regulation [6].

Avenues of research on non-coding RNAs (ncRNAs) in Archaea concerned the identification and characterization of small nucleolar RNAs, RNAs involved in rRNA modification [7] and RNAs involved in CRISPR-based immune systems [8]. In addition, several surveys for small regulatory ncRNAs in Archaea have been conducted [9]. *Sulfolobus solfataricus* (Sso) is a hyperthermophile that serves as a model organism for the crenarchaeal clade of Archaea. The genome of Sso contains a large number of mobile elements [10]. The genome has apparently undergone, and still undergoes, extensive rearrangements, which can be in part attributed to transposition events [10]. An RNomics approach identified 19 *cis*-antisense and 11 *trans*-encoded ncRNA candidates in Sso [11]. The majority of the *cis*-antisense RNAs were encoded opposite to transposase genes, suggesting that the RNAs could be involved in silencing of transposons. Moreover, high-throughput RNA sequencing (RNAseq) identified 185 *cis*-antisense and 125 *trans*-encoded ncRNA candidates in Sso [12]. It has been speculated that some of the *trans*-acting ncRNAs could regulate mRNAs by interacting with their 3′ UTRs [11], analogously to miRNAs in eukaryotes. However, the function, as well as the mechanism of these putative ncRNAs has

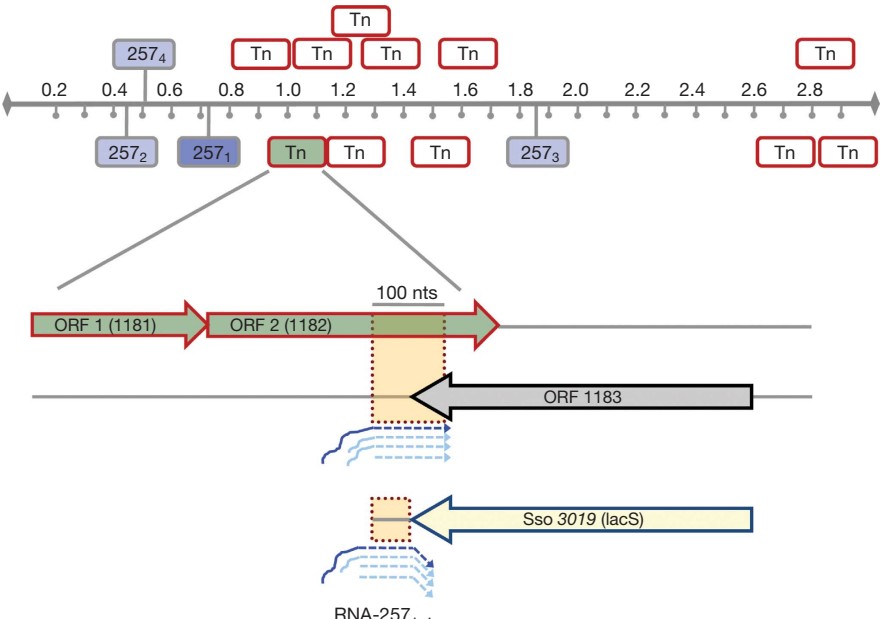

**Fig 1** | Several paralogs of the RNA-257$_1$ are present in the Sso genome. Location of RNA-257$_{1-4}$ (blue) and the 11 transposons (Tn; red boxes) in the Sso genome. The RNA-257$_1$ (dark blue) corresponds to the originally identified RNA-257 [11]. The transposon highlighted in green comprises Sso ORFs *1181* and *1182* (green arrow), which is opposite to Sso ORF *1183* (grey arrow). The RNAs-257$_{1-4}$ show complementarity to the 3′ end and the 3′ UTR of ORF *1183* (orange box). The *lacS-1183* 3′ UTR transcript is targeted by RNA-257$_{1-4}$ in *trans*.

remained unclear. With the exception of CRISPR-based immunity systems [13,14], there is so far only one report on antisense regulation in Archaea. In *Halobacterium salinarum*, an antisense RNA was shown to interact with the first 151 nucleotides of an early lytic phage transcript. This interaction results in cleavage of the mRNA and leads to removal of the ribosome-binding site, which renders the mRNA non-functional [15].

Here, we show that ncRNAs can regulate gene expression in the hyperthermophile Sso by interacting with complementary sequences present in the 3′ UTR of ORF *1183*. As a result, the mRNA is apparently destabilized, which is reminiscent to miRNA-mediated regulation in eukaryotes.

## RESULTS AND DISCUSSION
### Several paralogs of RNA-257 are present in Sso
Tang *et al* [11] have identified several putative *trans*-acting Sso ncRNAs, encompassing extended regions of complementarity with distinct mRNAs. In this study, we have focused on RNA-257, termed herein RNA-257$_1$. A Blast search (http://www-archbac. u-psud.fr/projects/sulfolobus/Blast_Search.html) revealed three paralogs of RNA-257$_1$, previously identified as RNA-107, RNA-91 and RNA-20 [11,12], which are renamed herein as RNA-257$_2$, RNA-257$_3$ and RNA-257$_4$ (supplementary Fig S1A online), respectively. RNA-257$_{1-4}$ are encoded in intergenic regions of the Sso genome (Fig 1), and all of them are transcribed [12]. The RNA-257$_1$ paralogs differ in length but posses a highly conserved core region (supplementary Fig S1A online). Further bioinformatic analyses disclosed a significant homology of these core regions with the distal coding region of a putative transposase gene of a transposon belonging to the ISC1904 family.

This transposon is present in eleven copies in the Sso genome (Fig 1). Most probably, RNA-257$_{1-4}$ are remnants of transposon rearrangements, whereby subtle nucleotide exchanges (supplementary Fig S1B online) created a promoter sequence, which led to the synthesis of RNA-257$_{1-4}$. One of the chromosomal copies of the transposons from which RNA-257$_{1-4}$ are apparently derived, is represented by Sso ORFs *1181* (putative resolvase gene) and *1182* (putative transposase gene; Fig 1). RNA-257$_{1-4}$, posses consensus-like promoter regions [16] (supplementary Fig S1B online). In contrast, only an imperfect promoter sequence is present upstream of the sequence corresponding to the conserved core sequence of RNA-257$_{1-4}$ in the distal part of the putative transposase genes of the respective transposons (supplementary Fig S1B online). According to an earlier report [16], the presence of the G at position four of box A (supplementary Fig S1B online) should prevent transcription. In agreement, RNAseq did not show a significant increase in reads corresponding to the 3′ end of ORF*1182* [12].

As the Sso ORFs *1182* and *1183* are convergently transcribed [12], the distal end of the *1182* transcript is complementary to the 3′ end as well as to the 3′ UTR of the *1183* transcript (Fig 1), encoding a putative phosphate transporter. Bioinformatic analyses revealed that ORF *1183* is conserved in other Sulfolobales (∼78% identity). However, the *1183* 3′ end and the 3′ UTR with complementarity to ORF *1182* and with RNA-257$_{1-4}$ is only present in Sso (supplementary Fig S1D online). The high homology of the core regions of RNA-257$_{1-4}$ to the distal end of *1182* mRNA, and thus the partial complementary to the 3′ end of *1183* mRNA, prompted us to ask whether regulation by antisense RNAs does occur in the hyperthermophile Sso.

## Phosphate-dependent abundance of RNA-257$_{1-4}$

To study the expression pattern of RNA-257$_1$ during different growth/ stress conditions, we first used a RNA-257$_1$-specific probe with complementarity to a less-conserved region (supplementary Fig S1A online). These studies revealed that the steady state levels of RNA-257$_1$ depend on the phosphate availability in the growth medium; the levels of RNA-257$_1$ were decreased under phosphate-limiting conditions (Fig 2A). In addition, northern blot analyses with probes specific for RNA-257$_2$, RNA-257$_3$ and RNA-257$_4$ indicated that their levels were likewise decreased under phosphate-limiting conditions (supplementary Fig S1C online). We can only speculate why expression of all RNA-257$_{1-4}$ genes is phosphate dependent. Inverted repeats have been described as important elements of phosphate-sensitive promoters in *Mycobacterium smegmatis* [17]. An inverted repeat preceding boxA is conserved in all four RNA-257$_{1-4}$ promoters (supplementary Fig S1B online). As the RNA-257$_{1-4}$ genes are most likely remnants of transposition events, it is conceivable that the inverted repeat, and thus the phosphate sensitivity of all four promoters, were generated during these events. At this junction, we did not further study transcriptional regulation of RNA-257$_{1-4}$. Instead, we asked whether the increased abundance of RNA-257$_{1-4}$ in full medium (plus phosphate) might correlate with a downregulation of *1183* mRNA encoding the putative phosphate transporter, that is, whether RNA-257$_{1-4}$ could be involved in negative antisense regulation of *1183* mRNA. Therefore, the levels of RNA-257$_1$ and *1183* mRNA were quantified in the presence of phosphate and under phosphate-limiting conditions employing qPCR. As shown in Fig 2B, an inverse correlation of the levels of RNA-257$_1$ and *1183* mRNA was observed under both conditions.

## Antisense regulation by *trans*-acting RNAs in Sso

As the levels of *1183* mRNA were decreased at elevated levels of RNA-257$_1$ (Fig 2B), these pilot studies provided a first hint for antisense regulation in a hyperthermophile. However, as ORF *1182* is convergently transcribed with *1183* mRNA, the above experiment did not distinguish whether *1183* mRNA is down-regulated in *trans* by RNA-257$_{1-4}$ or by a *cis*-antisense mechanism mediated by the full-length *1182* transcript. To address whether antisense regulation by RNA-257$_{1-4}$ can function in *trans*, the 3′ UTR of ORF *1183* was inserted downstream of the *lacS* gene (Fig 3A). The Sso strain PH1-16 was transformed with plasmid pMJ05 [18] (*lacS* with the authentic 3′ UTR) and plasmid pMJ05-1183 (*lacS* with the 3′ UTR of *1183*), respectively. In contrast to strain PH1-16(pMJ05), very low levels of ß-galactosidase activity were observed with strain PH1-16(pMJ05-1183) (supplementary Fig S2A online). As the transcripts are in both cases under the control of an arabinose-inducible promoter it was likely that the *lacS*-3′*UTR*-*1183* mRNA is targeted by RNA-257$_{1-4}$ in *trans*, and that the mRNA is subsequently degraded. Therefore, we next compared the *lacS* mRNA levels of strain PH1-16(pMJ05) and strain PH1-16(pMJ05-1183) grown in the presence of phosphate, using RT–PCR (supplementary Fig S2B online) and qPCR (Fig 3C). In contrast to strain PH1-16(pMJ05), no (supplementary Fig S2B online) or very low levels (Fig 3C) of *lacS*-3′*UTR*-*1183* mRNA were detected in strain PH1-16(pMJ05-1183), suggesting that *lacS*-3′*UTR*-*1183* mRNA is rapidly degraded *in vivo*.

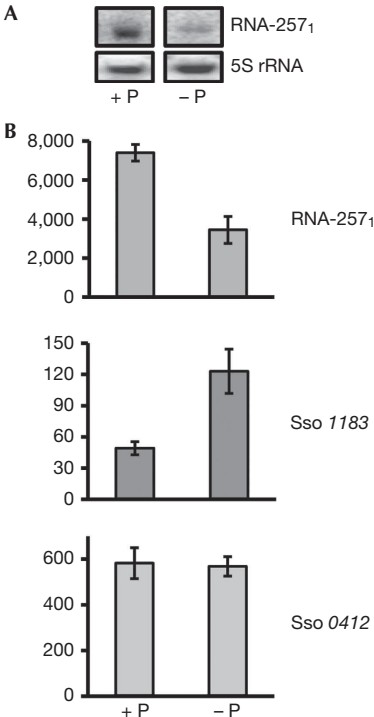

**Fig 2 | Inverse correlation of RNA-257$_1$ and *1183* mRNA levels in the presence and absence of phosphate. (A)** Northern blot analysis of RNA-257$_1$ levels with total RNA from Sso cells grown in either full medium ( + P; 280 mg/l KH$_2$PO$_4$) or under phosphate-limiting conditions ( − P). 5S rRNA was used as loading control. **(B)** qPCR analysis of RNA-257$_1$ and *1183* mRNA levels in the presence of phosphate ( + P) and under phosphate-limiting conditions ( − P). The Sso *0412* mRNA, encoding aIF2-γ, was used as a house-keeping endogenous control. The error bars represent s.d. from triplicate experiments. The numbers represent copies per 10 ng cDNA.

As multiple gene knockouts are not feasible in Sso, we could not delete the RNA-257$_{1-4}$ genes to unequivocally test whether these RNAs mediate antisense regulation of *1183* mRNA or of the *lacS*-3′UTR-*1183* transcript. Therefore, the putative RNA-257$_{1-4}$ base-pairing sequence within the *1183* 3′ UTR of the *lacS*-3′UTR-*1183* transcript was altered to reduce binding of RNA-257$_{1-4}$. First, 26 nucleotides of the base-pairing site (Fig 3B) within the 3′ UTR of the *1183* ORF were replaced in plasmid pMJ05-1183R26 by an unrelated sequence with the aim to weaken the interaction between RNA-257$_{1-4}$, and the *1183* 3′ UTR. Second, the entire *1183* 3′ UTR of the *lacS*-3′UTR-*1183* transcript was deleted in plasmid pMJ05-Δ to provide a mock control. When the cells were grown in the presence of phosphate, the replacement restored the *lacS* mRNA levels to some extent, whereas the entire deletion of the *1183* 3′ UTR within the *lacS*-3′UTR-*1183* transcript resulted in *lacS* mRNA levels comparable to that observed with plasmid pMJ05 (supplementary Fig S2B online; Fig 3C). In agreement, the strains PH1-16(pMJ05) and PH1-16(pMJ05-Δ) displayed equivalent β-galactosidase activities, whereas the strains PH1-16 (pMJ05-1183) and PH1-16(pMJ05-1183R26) displayed very low β-galactosidase activities (supplementary Fig S2A online).

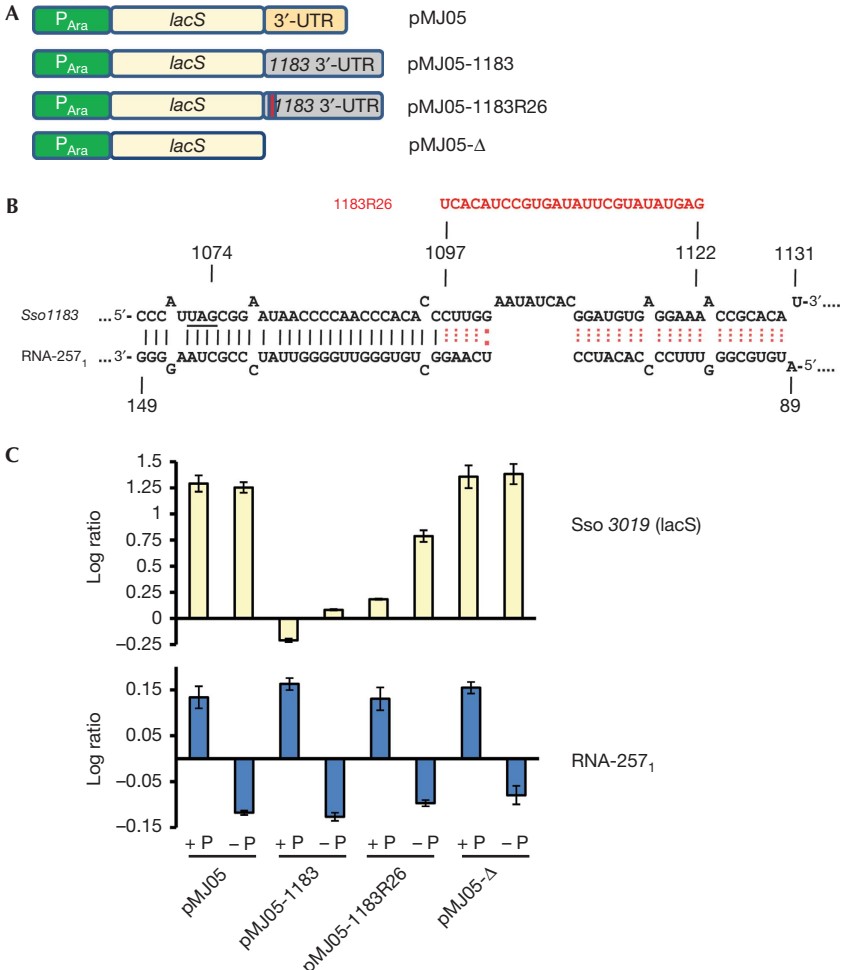

**Fig 3 | Replacement of the *lacS* 3′ UTR with the *1183* 3′ UTR results in destabilization of *lacS* mRNA. (A)** Schematic depiction of gene constructs in plasmids pMJ05, pMJ05-1183, pMJ05-1183R26 and pMJ05-Δ. The *lacS* gene is preceded by an arabinose-inducible promoter ($P_{Ara}$). The *lacS* 3′ UTR is replaced by the 3′ UTR of Sso *1183* in the pMJ05-1183 construct. In pMJ05-1183R26, 26 nt are replaced by an unrelated sequence (see **B**). In pMJ05-Δ the *lacS* gene is devoid of a 3′ UTR. **(B)** A part of the complementary region between RNA-$257_1$ and the 3′ UTR of ORF *1183* is depicted. Twenty-six nucleotides of the *1183* 3′ UTR were replaced by an unrelated sequence (1183R26 shown in red). **(C)** qPCR analysis of *lacS* and RNA-$257_1$ expression levels in the presence of phosphate ($+P$) and under phosphate-limiting conditions ($-P$). The transcript levels were normalized against that of Sso *0412*, encoding aIF2-γ. The numbers represent the log ratio normalized to expression values of Sso *0412*. The ratio of *lacS* mRNA: RNA-$257_1$ was calculated for strain PH1-16(pMJ05-1183) with 0.4 ($+P$) and 1.6 ($-P$) and for strain PH1-16(pMJ05-1183R26) with 1.3 ($+P$) and 7 ($-P$), respectively. The error bars represent s.d. from triplicate experiments.

The experiments shown in Fig 2 did not exclude the possibility that Sso *1183* levels could be as well regulated in a phosphate-dependent manner, that is, independently of RNA-$257_{1-4}$. To obtain additional support for the phosphate-dependent regulation of *1183* mRNA by RNA-$257_{1-4}$ (Fig 2), we tested whether the abundance of the *lacS* transcript in strains PH1-16(pMJ05), PH1-16(pMJ05-1183), PH1-16(pMJ05-1183R26) and PH1-16(pMJ05-Δ) is likewise phosphate dependent. Using qPCR, the *lacS* transcript abundance was compared in cells grown in full medium ($+P$) and under phosphate-limiting conditions ($-P$). As anticipated, the *lacS* (Fig 3C) and ß-galactosidase (supplementary Fig S2A online) levels were comparable under both conditions in strains PH1-16(pMJ05) and PH1-16(pMJ05-Δ).

In contrast, in the absence of phosphate, that is, at reduced levels of RNA-$257_1$, an increased abundance of both, the *lacS*-3′ UTR-*1183* and the *lacS*-3′UTR-1183R26 transcript was observed (Fig 3C). The abundance of the *lacS*-3′UTR-*1183*R26 transcript (*lacS*: RNA-$257_1$ ratio = 7) was higher than that of the *lacS*-3′UTR-*1183* transcript (*lacS*: RNA-$257_1$ ratio = 1.6), which was anticipated as the alteration within the *1183* 3′ UTR already attenuated the apparent regulation by RNA-$257_{1-4}$ (supplementary Fig S2B online; Fig 3C), which was obviously augmented by a simultaneous reduction of the RNA-$257_{1-4}$ levels. However, the ß-galactosidase activities obtained with plasmids pMJ05-1183 and pMJ05-1183R26 (supplementary Fig S2A online) were very low and not distinguishable when the cells were cultivated in the

presence or absence of phosphate. Apparently, there is enough RNA-$257_{1-4}$ under both conditions to drastically reduce the transcript levels of the corresponding *lacS* transcript (Fig 3D; supplementary Fig S2B online), which results in rather low ß-galactosidase activities (supplementary Fig S2A online), and in turn precludes a reasonable read out dependent on the levels of RNA-$257_{1-4}$. In summary, as a mutation of the base-pairing site as well as the phosphate-dependent modulation of the RNA-$257_{1-4}$ levels lead to increased or decreased levels of fusion constructs containing the *1183* 3′ UTR, these experiments supported the hypothesis that the *1183* 3′ UTR is targeted by the antisense RNAs in *trans*.

To verify the studies performed with the *lacS*-3′UTR-*1183* construct(s), the same sequence of the *1183* 3′part was fused to the coding region of the Sso Sm1 protein (ORF *6454*) in plasmid pMJ05 (supplementary Fig S3A online). In brief, with this fusion construct the same results as with the *lacS*-3′ UTR-*1183* construct were obtained. The same , inverse correlation between the abundance of the *6454*-3′UTR-*1183* transcript and that of RNA-$257_1$ were observed in the presence of phosphate and under phosphate-limiting conditions (supplementary Fig S3B online). In addition, under phosphate-limiting conditions the Sm1 protein levels of Sso strain PH1-16(pMJ05-6454-1183) were increased when compared with the same strain grown in the presence of phosphate (supplementary Fig S3C online).

## Degradation of *1183* mRNA by RNA-$257_1$

Duplex formation between a ncRNA and mRNA can create a cleavage site for dedicated riboendonucleases recognizing double-stranded RNA [18,19]. We therefore asked whether on addition of Sso S30 extracts a preformed duplex between RNA-$257_1$ and *1183* mRNA is subjected to accelerated degradation when compared with *1183* mRNA alone. We first tested whether duplex formation between RNA-$257_1$ and the 3′ UTR of *1183* mRNA occurs *in vitro* at 75 °C. An electrophoretic mobility shift assay (supplementary Fig S4A online) revealed that RNA-$257_1$ forms a duplex with the 3′ UTR of ORF *1183* mRNA, showing that base pairing between these RNAs is feasible. To further assess the stability of the duplex at high temperature, a melting curve analysis was performed. This analysis showed that the duplex between RNA-$257_1$ and the 3′ UTR of *1183* mRNA is stable below 75 °C, begins to melt at temperatures above 75 °C, and that full denaturation occurs at temperatures above 90 °C (supplementary Fig S4B online).

The *1183* mRNA and the preformed *1183* mRNA-RNA$257_1$ duplex, respectively, were added to Sso extracts. Samples were withdrawn after 0, 2, 4 and 6 min, transferred to a nylon membrane and degradation of *1183* mRNA was monitored using a probe specific for the 3′ end of the *1183* coding region (Fig 4B, lanes 1 and 2) as well as for the central part of *1183* mRNA (Fig 4B, lanes 3 and 4). The experiments were done in triplicate and the remaining *1183* mRNA was quantified. The degradation of the 3′ end of the *1183* coding region started after 4 min on addition of Sso extracts, whereas after 2 min 50% of the 3′ end of *1183* mRNA was degraded when the *1183* mRNA-RNA$257_1$ duplex was incubated with Sso extracts (Fig 4B, C). In contrast, during the 6-min time course no differences were observed in the stability for *1183* mRNA alone and of the *1183* mRNA-RNA$257_1$ duplex on addition of Sso

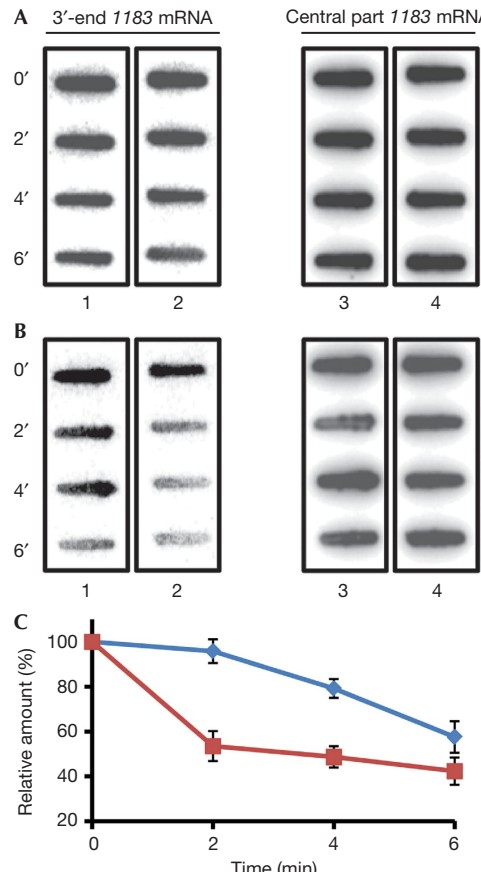

**Fig 4 |** Duplex formation between RNA-$257_1$ and *1183* mRNA augments degradation of *1183* mRNA *in vitro*. (**A**) Heat-inactivated Sso extracts and (**B**) 'active' S30 extracts were added to *1183* mRNA (lanes 1 and 3) or to the *1183* mRNA-RNA$257_1$ duplex (lane 2 and 4). After the indicated times in minutes, samples were withdrawn and the *1183* mRNA levels were determined with a probe specific for the 3′ end of *1183* mRNA (lane 1 and 2) and with a probe specific for the central part of *1183* mRNA (lane 3 and 4). The experiment was performed in triplicate. One representative experiment is shown. (**C**) Quantification of the results. The graph shows the relative amounts of the 3′ part of *1183* mRNA remaining in the absence of RNA-$257_1$ (blue) and when in duplex with RNA-$257_1$ (red) plotted as a function of time. The error bars represent s.d. from triplicate experiments.

extracts when the remaining *1183* mRNA was monitored with a probe directed against the central part of the mRNA (Fig 4B). The same set of experiments was performed with heat-inactivated S30 extracts to ascertain that degradation is dependent on components present in the Sso extracts. As shown in Fig 4A, no significant degradation of *1183* mRNA was observed on addition of heat-inactivated Sso extracts. Hence, we could recapitulate the apparent *in vivo* antisense regulation of *1183* mRNA and of the *lacS*-3′UTR-*1183* transcript *in vitro* using RNA-$257_1$.

## Concluding remarks

As noted above, we cannot completely dismiss a *cis*-acting mechanism or a direct, phosphate-dependent regulation of Sso

*1183* in the natural setting. However, as the phosphate-dependent modulation of the RNA-257$_{1-4}$ levels led to increased or decreased levels of *lacS/6454* fusion transcripts containing the *1183* 3′ UTR, we conclude that antisense regulation can function *in trans* in the hyperthermophile Sso. This raises the question as regarding RNases/factors involved in degradation of the target mRNA. In eukaryotes, two families of proteins have been shown to be required for RNAi pathways, the Dicer and Argonaute protein families [3]. Dicers belong to the RNase III family of riboendonucleases containing a PAZ domain, that bind and cleave double-stranded RNAs [3]. Argonaute proteins also possess endonuclease activity and require guide RNAs to cleave target mRNAs [3]. Besides the PAZ domain, Argonaute proteins contain an RNase H-like Piwi domain responsible for target RNA cleavage [3]. Proteins containing a Piwi domain on the basis of the Argonaute homologue of *Pyrococcus furiosus* [20] were only found in five strains belonging to the clade of Euryarchaeota [21]. In *Sulfolobus tokadaii*, a homologue of RNase HI [22], one of the three prokaryotic RNase H classes, was reported to cleave dsRNA [23]. In Sso, a gene encoding a homologue of RNase HII was identified [22]. It remains to be elucidated whether these proteins or proteins such as CRISPR crRNA-related Cas proteins [13,24] or as yet unidentified endoribonucleases are involved in degradation of Sso mRNAs targeted by antisense RNAs.

The most prominent group of Sso antisense RNAs identified by an RNomics approach is transcribed in the opposite orientation to transposase genes. Therefore, it has been speculated that these antisense RNAs regulate transposition events by inhibiting expression of transposase mRNA [11]. Similarly, the majority of eukaryal PIWI-interacting RNAs are antisense to transposon sequences, suggesting that they are involved in silencing of transposons [25]. In contrast, RNA-257$_{1-4}$ are sense to the transposase gene (ORF *1182*). The 3′ end of ORF *1183* and the 3′ UTR are not present in three other Sulfolobales (supplementary Fig S1D online). Thus, apparently only in Sso the transposon (ORF*1182*) inserted within ORF *1183*, thereby altering the 3′ end of this reading frame. This event apparently rendered *1183* mRNA vulnerable to antisense regulation by RNA-257$_{1-4}$ being descendants of the transposon ORF *1182*. Thus, RNA-257$_{1-4}$ could exemplify a means for the origin of trans-encoded regulatory RNAs.

## METHODS

**Strains and growth conditions.** *S. solfataricus* P2 was grown at 75 °C in Brock's medium containing either $KH_2PO_4$ to a final concentration of 280 mg/l (+P) or shifted to Brock's medium without $KH_2PO_4$ (−P; phosphate-limiting conditions). Electroporation of *S. solfataricus pyrEF lacS* mutant PH1-16 with plasmids pMJ05, pMJ05-1183, pMJ05-1183R26, pMJ05-Δ and pMJ05-6454-1183, and the isolation of transformants was performed as described [26]. See supplementary information online for further details.

**Northern blot analysis.** Total RNA from *S. solfataricus* strain P2 ($OD_{600} = 0.6$) grown in the presence and absence of $KH_2PO_4$, was isolated and separated on 8% polyacrylamide/urea gels and transferred to nylon membranes. After crosslinking, the membrane was incubated with either the [$^{32}$P]-5′-end labelled oligonucleotide 5′-GGCAGACCCGTTCATAC-3′ specific for RNA-257$_1$, the

oligonucleotide 5′-GGTGGTGCGTCATCAGATTAT-3′ specific for RNA-257$_3$ or the oligonucleotide 5′-GATTGTCTTACCAACC TTTC-3′ specific for RNA-257$_2$ and RNA-257$_4$. The 5S rRNA was probed with oligonucleotide 5′-CACTAACGTGAGCGGCTT AAC-3′ and served as loading control.

**RT–PCR and qPCR.** Total RNA from *S. solfataricus* strains P2, PH1-16(pMJ05), PH1-16(pMJ05-1183), PH1-16(pMJ05-1183R26), PH1-16(pMJ05-Δ), PH1-16(pMJ05-6454-1183), grown in either full medium (phosphate present) or under phosphate-limiting conditions was isolated and complementary DNA was synthesized using random hexamer oligos (Fermentas) and SuperScriptIII Reverse Transcriptase (Invitrogen). See supplementary information online for further details.

***In vitro* RNA degradation assay.** S30 extracts of *Sulfolobus solfataricus* P2 were either pre-incubated for 20 min at 75 °C or heat-inactivated for 20 min at 100 °C. To achieve duplex formation, full-length *1183* mRNA (0.16 μM) was incubated with RNA-257$_1$ (0.48 μM) followed by addition of either the 'active' or the heat-inactivated extracts (0.3 μg protein/μl) and incubated at 75 °C. Samples were withdrawn at 0, 2, 4 and 6 min, the reaction was terminated by addition of 1 mM EDTA and an equal amount of 2 × RNA loading dye, and then transferred to a nitrocellulose membrane. After crosslinking either the [$^{32}$P] 5′-end labelled oligonucleotide 5′-GAAATCCCATGAAGCCCAAAC-3′ (complementary to nucleotides 979–999 of *1183* mRNA) or the [$^{32}$P] 5′-end labelled oligonucleotide 5′-CTGTGTTCAGCCATTATCG-3′ (complementary to nucleotides 410–428 (central part) of *1183* mRNA) was added to detect *1183* mRNA. See supplementary information online for further details.

ACKNOWLEDGEMENTS
This work was supported by grant 22888 from the Austrian Science Fund (FWF) to U.B. We thank R. Garrett and V. Sedlyarov for helpful discussions and P. Blum for materials.

Author contributions: B.M., D.H. and U.B. designed experiments; B.M., S.M. and A.M. performed experiments; B.M. and U.B. wrote the manuscript.

CONFLICT OF INTEREST
The authors declare that they have no conflict of interest.

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
