## [Review Process File · EMBO Reports]

Manuscript EMBOR-2012-36589

Antisense regulation by transposon derived RNAs in the hyperthermophilic archaeon *Sulfolobus solfataricus*

Birgit Märtens, Salim Manoharadas, David Hasenöhrl, Andrea Manica and Udo Bläsi

Corresponding author: Udo Bläsi, Max F. Perutz Laboratories

Review timeline:	Submission date:	04 September 2012
	Editorial Decision:	11 October 2012
	Revision received:	10 January 2013
	Editorial Decision:	25 January 2013
	Revision received:	08 March 2013
	Editorial Decision:	15 March 2013
	Accepted:	18 March 2013

Editors: Alejandra Clark / Esther Schnapp

Transaction Report:

1st Editorial Decision

11 October 2012

Thank you for the submission of your research manuscript to EMBO reports. Please accept my sincere apologies for the unusual delay in the review process of your manuscript that was due to the fact that we have just received the full set of reports on it, which I copy below.

As you will see, although all referees find the study potentially interesting they raise a number of significant concerns, most of which need to be experimentally addressed. All referees stress that further experiments are required to demonstrate that RNA-257-1 is a phosphate-dependent regulatory RNA. Specifically, they suggest that the lacS reporter assay should be performed under conditions of phosphate abundance and starvation. In addition, since RNA-257-1 is derived from transposons, and Sso is atypical in that it contains a large number of transposable elements, all reviewers indicate that the generality of the described mechanism of post-transcriptional regulation by a trans-acting anti-sense RNA should be at least discussed, or better, if possible, experimentally addressed. For instance, referee #3 suggests that a related *Sulfolobus* species that lack such large number of transposons should be investigated for the presence of a post-transcriptional regulation mechanism. Importantly, referees #2 and #3 are concerned that the current evidence based on a lacS reporter assay does not directly demonstrate anti-sense regulation by the trans-acting RNA *in vivo*, and suggest that further experiments to distinguish between cis- and trans-acting mechanisms should be performed. Finally, all reviewers point to additional issues including technical concerns and

further controls.

Given the potential interest of the novel findings and considering that all referees provide constructive suggestions on how to move the study forward, I would like to give you the opportunity to revise the manuscript, with the understanding that the main referees concerns, including those mentioned above have to be addressed and that acceptance of the manuscript would entail a second round of review. I would like to point out that it is EMBO reports policy to allow a single round of revision only and that thus acceptance or rejection of the manuscript will depend on the outcome of the next final round of peer-review.

Revised manuscripts should be submitted within three months of a request for revision; they will otherwise be treated as new submissions. If you feel that this period is insufficient to address all the referee concerns I can potentially extend this period slightly. Also, the length of the revised manuscript should not exceed 30,000 characters (including all text and spaces), and not more than 5 main plus 5 supplementary figures may be included. Should you find the length constraints to be a problem, you may consider including some peripheral data and/or materials and methods in the form of Supplementary information. However, materials and methods essential for the understanding of the key experiments must be described in the main body of the text and may not be moved to the supplemental information.

We have also started encouraging authors to submit the raw data for western blots (i.e. original scans) to our editorial office. These data will be published online as part of the supplementary information. This is voluntary at the moment, but if you agree that this would be useful for readers I would like to invite you to supply these files when submitting the revised version of your study.

As part of the EMBO publication's Transparent Editorial Process, EMBO reports publishes online a Review Process File to accompany accepted manuscripts. This File will be published in conjunction with your paper and will include the referee reports, your point-by-point response and all pertinent correspondence relating to the manuscript.

We also welcome the submission of cover suggestions or motifs that might be used by our Graphics Illustrator in designing a cover.

I look forward to seeing a revised form of your manuscript when it is ready. Should you in the meantime have any questions, please do not hesitate to contact me.

REFeree REPORTS:

Referee #1

This paper from the Bläsi lab reports the novel finding of antisense regulation by a prokaryotic small RNA that targets the 3' UTR of an mRNA. Small RNAs that regulate mRNAs by base pairing mechanisms are being investigated intensively in eubacteria and eukaryotes, and these studies have identified many unexpected similarities, for example, that most small RNAs from these kingdoms may recognize their targets by short seed pairing. However, they have also led to the general notion that prokaryotic/eubacterial and eukaryotic small RNAs generally differ in their preference for mRNA binding: the prokaryotic sRNAs analysed so far operate in the mRNA 5' region whereas the eukaryotic microRNAs preferentially target the 3' region of mRNA.

The present manuscript is the first to identify trans action in archaea and suggests that similar to eukaryotes, archaeal small RNAs may generally target the 3' UTR of mRNAs. Recent RNA deep sequencing studies have suggested the existence of hundreds of small RNAs in archaea, but our knowledge of the molecular mechanisms of RNA-based gene regulation in this kingdom has been minimal as compared to eubacteria and eukaryotes. As such the present study is a

major advance in the field of gene regulation. Most of the experiments are well-executed and most of the presented data is clear. Prior to publication in EMBO Reports, I would strongly recommend some revisions to address the following points.

Major issues:

1. The presented evidence that RNA-257/1 binds the 3' UTR of the 1183 mRNA is cumulative, i.e. represents the sum of indirect results which include *in silico* prediction of a long antisense interaction, disruption of this putative interaction *in vivo*, and complex formation (gel shifts experiments) *in vitro*. The lack of a good experimental system that would not allow the authors to introduce compensatory base pair changes to validate base pairing *in vivo*, which is the gold standard in eubacteria. However, the authors present in Figure 4 a lysate-based RNA stability assay that would lend itself as a surrogate system to study the effects of mutations in the predicted RNA duplex and seek restoration of base pairing by compensatory mutations in the sRNA and the target.
2. The *in vitro* RNA stability results presented in Figure 4 are difficult to interpret with respect to reproducibility. The experiments should be performed in triplicates, and RNA stability should be presented as graphs to be able to calculate RNA half-lives and judge the variation in the assay. In addition, as suggested above, informative mutant versions of the interacting RNAs should be included.
3. Figure 3: Is the downregulation of the 3'UTR reporters phosphate-dependent? In other words, would phosphate-starvation repressing the small RNAs elevate the expression of the pJM105-1183 and pJM105-1183-R26 reporters? This would be important to convince the reader that these reporters fully recapitulate the small RNA-mediated repression of the native 1183 mRNA.

Minor issues:

4. Page 3, Introduction: I recommend this part of the manuscript be used to make clear that targeting the 5' UTR versus 3' UTR has been a major difference between prokaryotes and eukaryotes, for the general reader to appreciate the key novelty of the present work.
5. Page 5, middle part: Does the listing of other families of transposon-derived small RNA genes really contribute to an understanding of the ncRNA-257 family in the context of the current paper?
6. Page 6, line 15, "... were increased and decreased in the presence and absence of phosphate": Since the authors examine only two conditions, it is hard to say whether there was an increase AND a decrease. If phosphate-replete is the reference condition, it will do to say that a decrease was observed in the absence of phosphate.
7. Page 8, last paragraph of main text: After presenting the results for the RNA stability, the text jumps to a discussion of nucleases that might be involved in the regulation. It would be helpful, however, to briefly summarize the results of this work, and critically reflect the evidence for and potential generality of the reported regulation in the 3' UTR.
8. Figure 1 requires improvement. It is difficult to understand where the RNA-257 copies are located (are they really 300kb long?) and whether or not RNA-257-1 is genetically linked with the green Tn.

Referee #2

SUMMARY:

1. This report has the single key claim that archaea have mechanisms of small RNA-based regulation, where the small RNA targets the 3'UTR of its substrate mRNA, leading to a degradation.
2. This would be a novel and significant finding, since very little is known on small RNA-based regulation in archaea.

3. The finding is of general interest to a broader public, especially if compared to the more established small RNA-based regulation in eukaryotes and bacteria.
4. The experiments need further confirmation and controls to fully justify the conclusions as outlined below.

FULL REPORT:

Background/ General Interest:

The regulation of gene expression via small noncoding RNAs is an exciting and competitive research field with far-reaching implications. Whereas significant progress has been made to understand the respective systems in eukaryotes and bacteria, research on regulatory, non-coding RNA in archaea is at the very beginning.

Summary:

The present report claims to uncover the first example in archaea of a small non-coding RNA to target the 3'UTR of an mRNA in trans by extensive base-pairing and to regulate its abundance. In contrast to bacteria, the 3'UTR is the typical target for eukaryotic microRNAs to regulate their targets. Interestingly, both the non-coding RNA (RNA-257-1) and its target (the 3'UTR of Sso1183, encoding a putative phosphate transporter) are derived from an archaeal DNA transposon.

Significance:

- Hence, if 3' UTR targeting should turn out to be a primary and more widespread mechanism in archaea, this would be a highly significant finding.
- Furthermore, the transposon-based origin of the regulation could be worthwhile to be discussed in an evolutionary context of how small RNA-based regulatory mechanisms might emerge and spread.

Comments to the authors:

General:

Before publication, however, the authors should corroborate their findings that RNA-257-1 is a phosphate-dependent regulatory RNA that mediates the degradation of its target RNA. Also, the author need to provide the reader with a more comprehensive discussion of how unique/ significant/ widespread the described regulatory mechanism is in archaea and how they judge the transposon-derived origin of the RNA sequences.

Major individual points:

1. It is unclear to which degree the described effects are phosphate-specific. The authors should try and demonstrate this more thoroughly.

1.1 They should make available the results of the tested growth/ stress conditions as part of the Supplement to illustrate that phosphate levels are indeed the primary cause of the different RNA-257-1 levels. Add phosphate concentrations to this Supplemental Figure (and to Figure 2).

1.2 Include titration experiments at several intermediate phosphate concentrations and analyze RNA levels by qPCR as in Figure 2B. Show phosphate response curves for RNA-257-1, Sso1183, Sso0412 and include the Sso3019 (*lacS*) constructs pMJ05-1183-R26 and pMJ05 (and/or pMJ05-delta). If possible, also include RNA-257-2, RNA-257-3, and RNA-257-4. It would be surprising if the transcription of all four versions of RNA-257 were phosphate-dependent. The authors should explain to the reader why they focused exclusively on RNA257-1. Furthermore, it is crucial for the proposed regulation 'in trans' to demonstrate that the amount of Sso3019 RNA as derived from the pMJ05-1183-R26 plasmid (Figure 3 and Point 2.2) is phosphate-dependent.

1.3 If feasible, add phosphate to the medium and follow RNA levels by qPCR in a time-course experiment. This might demonstrate that levels of RNA257-1 rise BEFORE levels of Sso1183 drop and it might help to establish cause and effect. In any case, the authors are encouraged to explain to the reader why they do not consider the alternative hypothesis that Sso1183 levels are regulated independently of RNA-257-1 and simply have an effect on RNA-257-1 levels as a consequence of the base-pairing (e.g. use the amounts of RNA in Figure 2B as an argument).

2. The experiment in Figure 3 (transfer of the Sso1183 3'UTR to a lacS reporter) is an important and crucial experiment to demonstrate that RNA257-1 can work in trans. The experiment should include additional controls.

2.1 It may be problematic to introduce transposon-derived 1182 DNA on the pMJ05 plasmid. This could potentially trigger some host transposon defense mechanisms or recombination events that prevent the lacS fusion RNA to be properly transcribed in the first place. This would provide an alternative explanation for the complete absence of evidence on the protein and RNA levels for pMJ05-1183. Is there a way to check for plasmid integrity or to control for such effects?

2.2 The R26 mutation of pMJ05-1183 apparently attenuates regulation by RNA-257-1, such that pMJ05-1183-R26 RNA gets detectable by RT PCR (Figure 3C), but the low amount of pMJ05-1183-R26 RNA indicates that is still under control of RNA-257-1 due to partial complementarity. If this is the case, levels of pMJ05-1183-R26 RNA should be phosphate dependent (see the crucial experiment suggested under Point 1.2). The authors should demonstrate phosphate dependence of pMJ05-1183-R26 RNA to support their claim that phosphate-dependent regulation of Sso1183 RNA occurs via RNA-257-1.

3. The rationale and motivation for the in vitro experiments in Figure 4 are not very clear. At best, one can conclude that *S. solfataricus* contains double strand-specific RNases, but if the presented experiments recapitulate the physiological mechanism of Sso1183 RNA degradation in any way remains doubtful. It may be more helpful to move Figure 4 to the Supplement and rather investigate first how pMJ05-1183-R26 RNA (Figure 3C) is degraded in vivo. To this aim one could do Northern blots to see how much of pMJ05-1183-R26 RNA is full-length and whether there are specific degradation intermediates (e.g. whether the complementary part of 3'UTR is degraded preferentially). Alternatively, this could be done by qPCR, targeting different regions of the mRNA.

4. To stimulate the interest of a more generally interested audience, the authors should provide a more comprehensive discussion of how unique/ significant/ widespread the described regulatory mechanism might be in archaea and what are the obvious DIFFERENCES to the eukaryotic microRNA mechanisms. Which obvious questions need to be addressed next? How do the authors see the transposon-derived origin of RNA-257-1? Is it a problem for the generalization of the mechanism?

Referee #3

This study investigates changes in gene expression of a putative phosphate transporter (ORF 1183) likely in response to one or more transposon-derived transcripts RNA-2571-4 and conditions of low phosphate in *Sulfolobus solfataricus* (Sso). A main claim of the paper is that this is the first example of anti-sense regulation by trans-acting RNAs in a hyperthermophilic archaeon. However, as described below the evidence that the control is exerted in trans rather than cis is not convincing, the generality of the observation is unclear/suspect, and the analysis is quite preliminary in nature.

1. The significance of the work from a broader perspective is diminished by the finding (not clarified in the abstract) that the transcripts that purportedly regulate stability of the mRNA are very

likely transposon derived. *Sso* is very atypical in that it has an extraordinary number of active transposable elements that result in large-scale genomic changes. Thus, the phenomenon under study may be a rare example of post-transcriptional control by non-coding RNAs as the result of recent events rather than a more deep rooted and pervasive gene expression regulation program commonly employed by archaea. For example, in related *Sulfolobus* species that lack large numbers of transposons is there evidence of a similar and conserved post-transcriptional regulation mechanism?

2. As the authors describe, the effect observed on ORF 1183 mRNA appears to result from convergent transcripts from immediately adjacent ORF 1182. The data presented does not exclude the possibility that the observed regulation is brought about via *cis* rather than (claimed) *trans* regulation. The observation that the effect can be artificially recapitulated in *trans* (Fig 3) with a reporter system does not speak to if the effect acts in *trans* physiologically. As the authors concede, a more definitive test (systematic gene knock-out of each species of RNA-2571-4) is currently not feasible.

3. No work was reported to address if the effects on gene expression observed under conditions of phosphate starvation or abundance occurs *in vivo* by transcriptional (rather than post-transcriptional) mechanisms differentially operating on the two convergent promoters that separately regulate expression of RNA-2571 or ORF1183 mRNA. Moreover, the *lacS* reporter assay should be performed under conditions of both phosphate abundance and starvation to address if the system recapitulates the regulation observed *in vivo* (Fig 2). Moreover, Northern probes that detect all of the 4 size forms would address if each RNA responds to phosphate limitation as does the *cis*-encoded RNA-2571 form that was specifically examined.

4. In Fig 4, the stability of 1183 mRNA was found to be higher if it was not hybridized to RNA-2571 duplex when exposed to cell-free extract indicating a possible mechanism of action through degradation by an unknown double-strand specific RNA endonuclease. The analysis would have been more informative if the decay of end-labeled or (best) body labeled full-length mRNA was carried out rather than the indirect method (Northern analysis probing specific region of the mRNA). The expected mRNA decay intermediates that would have been possible to observe using this approach and a characterization of these breakdown products could be informative with regard to differential stability observed between free and duplexed mRNA.

1st Revision - authors' response

10 January 2013

Editor:

1. All referees stress that further experiments are required to demonstrate that RNA-257-1 is a phosphate-dependent regulatory RNA. Specifically, they suggest that the *lacS* reporter assay should be performed under conditions of phosphate abundance and starvation.

Experiments addressing these questions are now shown in Fig 3D.

2. In addition, since RNA-257-1 is derived from transposons, and *Sso* is atypical in that it contains a large number of transposable elements, all reviewers indicate that the generality of the described mechanism of post-transcriptional regulation by a *trans*-acting anti-sense RNA should be at least discussed, or better, if possible, experimentally addressed. For instance, referee #3 suggests that a related *Sulfolobus* species that lack such large number of transposons should be investigated for the presence of a post-transcriptional regulation mechanism.

We mention on page 5, discuss on page 10, and show in Fig S1D that the targeting sequence of RNA-257₁₋₄, i.e. the 3' end and the 3'UTR of 1183 mRNA is only present in *Sso* but not in three other *Sulfolobales*. No ncRNAs have so far been identified in other *Sulfolobus* species. Our current knowledge on (putative) ncRNAs is therefore restricted to *S. solfataricus*. At present, studies in a related *Sulfolobus* species would have to start from scratch and can hardly be addressed in the timeframe given for the revision. Nevertheless, we still think that the story in *Sso* is interesting on its own, as it also provides an example of

how regulatory ncRNAs can possibly evolve. We discussed this on page ten and changed the title as to the origin of the antisense RNAs.

3. Importantly, referees #2 and #3 are concerned that the current evidence based on a lacS reporter assay does not directly demonstrate anti-sense regulation by the trans-acting RNA in vivo, and suggest that further experiments to distinguish between cis- and trans-acting mechanisms should be performed.

To verify the findings with the *lacS*-3'UTR-1183 construct, we have performed an additional study with a similar 6454-3'UTR-1183 construct. ORF 6454 encodes the Sso Sm1 protein. In this case we were able to demonstrate a correlation between the intracellular levels of Sm1 protein, 6454 mRNA and RNA-257₁ (please see ms, page 8 and Fig S3). As for the *lacS*-3'UTR-1183 RNA, a reasonable assumption is that RNA-257₁ acts in *trans* to bring about the observed regulatory effect.

4. Finally, all reviewers point to additional issues including technical concerns and further controls.

Please see at the respective comments of the reviewers.

Referee#1:

1. The presented evidence that RNA-257/1 binds the 3' UTR of the 1183 mRNA is cumulative, i.e. represents the sum of indirect results which include in silico prediction of a long antisense interaction, disruption of this putative interaction in vivo, and complex formation (gel shifts experiments) in vitro. The lack of a good experimental system that would not allow the authors to introduce compensatory base pair changes to validate base pairing in vivo, which is the gold standard in eubacteria. However, the authors present in Figure 4 a lysate-based RNA stability assay that would lend itself as a surrogate system to study the effects of mutations in the predicted RNA duplex and seek restoration of base pairing by compensatory mutations in the sRNA and the target.

We have introduced deletions (25 bases) in 1183 mRNA as well as in RNA-257₁. However, due to the long complementarity between both RNAs, we still observed duplex formation in band shift assays when we incubated wt 1183 RNA with mutated RNA-257₁ or *vice versa* when mutated 1183 RNA was incubated with wt RNA -257₁. Presumably as a result, we did not observe significant differences in the stability of the 3' end of 1183 mRNA when the respective 1183:RNA-257₁ duplexes were incubated in Sso extracts. Another complication towards observing differences in "stability" with the mutant RNAs is the short half life of the 3' end of 1183 mRNA (see Fig. 4C) when in duplex with RNA-257₁.

This issue is also obvious from our studies with plasmid pMJ05-1183-R26 (see Fig 3C). Although 26 nt were replaced in the construct to diminish binding of RNA-257₁₋₄, we still observe reduced levels of *lacS*-3'UTR-1183R26 RNA, when compared with *lacS* (with its authentic 3'UTR) or *lacS*Δ (Fig. 3C) indicating that RNA-257₁₋₄ can still base-pair with *lacS*-3'UTR-1183R26.

2. The in vitro RNA stability results presented in Figure 4 are difficult to interpret with respect to reproducibility. The experiments should be performed in triplicates, and RNA stability should be presented as graphs to be able to calculate RNA half-lives and judge the variation in the assay. In addition, as suggested above, informative mutant versions of the interacting RNAs should be included.

We performed the experiment with the 1183 mRNA +/- RNA-257 in triplicate and quantified the blots with image quant. The resulting graph is now shown in Fig 4C. To verify these data we also performed similar assays with body-labeled *lacS*-3'UTR-1183 RNA (see Fig S5).

3. Figure 3: Is the downregulation of the 3'UTR reporters phosphate-dependent? In other words, would phosphate-starvation repressing the small RNAs elevate the expression of the pJM105-1183 and pJM105-1183-R26 reporters? This would be important to convince the reader that these reporters fully recapitulate the small RNA-mediated repression of the native 1183 mRNA.

As suggested, we repeated the experiment and grew the *Sso* cells expressing the *lacS*-3'UTR-1183 and *lacS*-3'UTR-1183R26 transcripts in full medium (+P) and under phosphate limiting conditions (-P). Using RT-PCR the *lacS* mRNA levels were determined (Fig. 3D). For both "*lacS*- constructs" we observed elevated levels during phosphate limiting conditions, i.e. under conditions when the levels of RNA-257₁₋₍₄₎ are decreased (see Fig. 3D and Fig S1C)

Minor issues:

4. Page 3, Introduction: I recommend this part of the manuscript be used to make clear that targeting the 5' UTR versus 3' UTR has been a major difference between prokaryotes and eukaryotes, for the general reader to appreciate the key novelty of the present work.

To make this more explicit one sentence was added at the end of the first paragraph on page 3.

5. Page 5, middle part: Does the listing of other families of transposon-derived small RNA genes really contribute to an understanding of the ncRNA-257 family in the context of the current paper?

As suggested and given the space constraints, we have eliminated the information on RNA-75 and RNA-108.

6. Page 6, line 15, "... were increased and decreased in the presence and absence of phosphate": Since the authors examine only two conditions, it is hard to say whether there was an increase AND a decrease. If phosphate-replete is the reference condition, it will do to say that a decrease was observed in the absence of phosphate.

We have changed the language according to the suggestion of the referee by stating were decreased under phosphate limiting conditions...

7. Page 8, last paragraph of main text: After presenting the results for the RNA stability, the text jumps to a discussion of nucleases that might be involved in the regulation. It would be helpful, however, to briefly summarize the results of this work, and critically reflect the evidence for and potential generality of the reported regulation in the 3' UTR.

We have done so and have also added more text on page 10 with regard to the possible origin of ncRNAs.

8. Figure 1 requires improvement. It is difficult to understand where the RNA-257 copies are located (are they really 300kb long?) and whether or not RNA-257-1 is genetically linked with the green Tn.

Figure 1 was modified.

Referee #2:

1. It is unclear to which degree the described effects are phosphate-specific. The authors should try and demonstrate this more thoroughly.

Please see our response to comment 3 of referee #1 and below.

1.1 They should make available the results of the tested growth/ stress conditions as part of the Supplement to illustrate that phosphate levels are indeed the primary cause of the different

RNA-257-1 levels. Add phosphate concentrations to this Supplemental Figure (and to Figure 2).

We tested the abundance of RNA-257₁ in logarithmic, in stationary phase and during nutrient starvation. Under these conditions we did not observe any significant differences in full medium (+P). As these blots only show an equal abundance and the number of suppl. Figs. is limited to 5, we preferred to rather show the phosphate dependent abundance of RNA-257₂₋₄ in Fig S1C.

The KH₂PO₄ concentration in full medium was 280 mg/l. The cells were then diluted in medium without KH₂PO₄. We have added more information on "phosphate limiting conditions in Suppl. Information and added the concentration of KH₂PO₄ in the legend to Fig 2.

1.2 Include titration experiments at several intermediate phosphate concentrations and analyze RNA levels by qPCR as in Figure 2B. Show phosphate response curves for RNA-257-1, Sso1183, Sso0412 and include the Sso3019 (*lacS*) constructs pMJ05-1183-R26 and pMJ05 (and/or pMJ05-delta). If possible, also include RNA-257-2, RNA-257-3, and RNA-257-4. It would be surprising if the transcription of all four versions of RNA-257 were phosphate-dependent. The authors should explain to the reader why they focused exclusively on RNA257-1. Furthermore, it is crucial for the proposed regulation 'in trans' to demonstrate that the amount of Sso3019 RNA as derived from the pMJ05-1183-R26 plasmid (Figure 3 and Point 2.2) is phosphate-dependent.

We have only determined the levels of RNA-257₁₋₄ in full Brock's medium and Brocks medium depleted for phosphate. We used RNA-257₃ and RNA-257_{2/4} specific probes to test if the paralogues RNAs are also down-regulated under phosphate limiting conditions. As shown in Fig S1C, the steady state levels of RNA-257₂, RNA-257₃ and RNA-257₄ are likewise decreased under these conditions.

As suggested we show in Fig. 3D that the amount of of Sso 3019 mRNA is phosphate dependent when expressed in strains PH1-16(pMJ05-1183) and PH1-16 (pMJ05-1183R26), respectively. While the level of Sso 3019 mRNA was increased during phosphate limitation, the levels of RNA-257₁ were decreased (Fig 3D). In contrast, the abundance of the two control mRNAs 0412 and 0764 was the same regardless whether the cells were grown in full medium or in medium depleted for phosphate (Fig 3D). We focused on RNA-257₁ being one representative of RNA-257₁₋₄.

1.3 If feasible, add phosphate to the medium and follow RNA levels by qPCR in a time-course experiment. This might demonstrate that levels of RNA257-1 rise BEFORE levels of Sso1183 drop and it might help to establish cause and effect. In any case, the authors are encouraged to explain to the reader why they do not consider the alternative hypothesis that Sso1183 levels are regulated independently of RNA-257-1 and simply have an effect on RNA-257-1 levels as a consequence of the base-pairing (e.g. use the amounts of RNA in Figure 2B as an argument).

In the revised version we have made these concerns apparent for the reader on page 6. To verify the findings obtained with 1183, we transplanted the 1183 3'UTR to the 3'end of *lacS*, expression of which (*lacS* only) is independent of phosphate (see Fig 2B). Only when the 1183 3'UTR was fused to *lacS*, the abundance of the *lacS*-3'UTR-1183 became phosphate dependent (Fig 3D).

Moreover, to verify the findings with the *lacS*-3'UTR-1183 construct, we have performed an additional study with a similar 6454-3'UTR-1183 construct. ORF 6454 encodes the Sso Sm1 protein. In this case we were able to demonstrate a correlation between the intracellular levels of Sm1 protein, 6454 mRNA and RNA-257₁ (please see ms., page 8 and Fig S3).

2. The experiment in Figure 3 (transfer of the Sso1183 3'UTR to a *lacS* reporter) is an important and crucial experiment to demonstrate that RNA257-1 can work in trans. The experiment should include additional controls.

2.1 It may be problematic to introduce transposon-derived 1182 DNA on the pMJ05 plasmid. This could potentially trigger some host transposon defense mechanisms or recombination events that prevent the lacS fusion RNA to be properly transcribed in the first place. This would provide an alternative explanation for the complete absence of evidence on the protein and RNA levels for pMJ05-1183. Is there a way to check for plasmid integrity or to control for such effects?

We introduced the 3' UTR of 1183 that is partially complementary to the transposon ORF 1182. We confirmed the presence / integration of the plasmid after the transformation of Sso cells growing from single colonies, which we used for this study. To make this clear we have added more text to Suppl. Information (see: Transformation of Sso).

We could confirm the transcription, as low RNA levels of *lacS* mRNA became detectable if we used more cDNA (500 ng) for the PCR; the experiment is now shown in Fig. S2C.

2.2 The R26 mutation of pMJ05-1183 apparently attenuates regulation by RNA-257-1, such that pMJ05-1183-R26 RNA gets detectable by RT PCR (Figure 3C), but the low amount of pMJ05-1183-R26 RNA indicates that is still under control of RNA-257-1 due to partial complementarity. If this is the case, levels of pMJ05-1183-R26 RNA should be phosphate dependent (see the crucial experiment suggested under Point 1.2). The authors should demonstrate phosphate dependence of pMJ05-1183-R26 RNA to support their claim that phosphate-dependent regulation of Sso1183 RNA occurs via RNA-257-1.

We have performed the suggested experiment (see Fig 3D). Sso cells expressing the *lacS*-3'UTR-1183 and *lacS*-3'UTR-1183R26 transcripts were grown in full media (+P) and under phosphate limiting conditions (-P) and the *lacS* mRNA levels were determined using RT-PCR. For both constructs we see elevated levels of *lacS* during phosphate limitation. See text, page 7 and 8.

3. The rationale and motivation for the in vitro experiments in Figure 4 are not very clear. At best, one can conclude that S. solfataricus contains double strand-specific RNases, but if the presented experiments recapitulate the physiological mechanism of Sso1183 RNA degradation in any way remains doubtful. It may be more helpful to move Figure 4 to the Supplement and rather investigate first how pMJ05-1183-R26 RNA (Figure 3C) is degraded in vivo. To this aim one could do Northern blots to see how much of pMJ05-1183-R26 RNA is full-length and whether there are specific degradation intermediates (e.g. whether the complementary part of 3'UTR is degraded preferentially). Alternatively, this could be done by qPCR, targeting different regions of the mRNA.

Fig 4: Although we observed a correlation between the levels of RNA-257 and the "reporter RNAs" we sought a means to show that binding of RNA-257 to the complementary stretch present at the 3' end and in the 3'UTR results in faster degradation of this part.

We have performed an additional "in vitro degradation experiment" (see Fig. S5) with a body-labeled 3' fragment of the *lacS*-3'UTR-1183 transcript (see referee #3, point 4; Fig S5). This experiment revealed that the part of *lacS*-3'UTR-1183 complementary to RNA257₁ is faster degraded in vitro than the *lacS*-3'UTR-1183 in the absence of RNA-257₁ (see Fig S5). However, we could not discern specific degradation intermediates under these conditions.

4. To stimulate the interest of a more generally interested audience, the authors should provide a more comprehensive discussion of how unique/ significant/ widespread the described regulatory mechanism might be in archaea and what are the obvious DIFFERENCES to the eukaryotic microRNA mechanisms. Which obvious questions need to be addressed next? How do the authors see the transposon-derived origin of RNA-257-1? Is it a problem for the generalization of the mechanism?

Here, we only aimed at providing evidence that antisense regulation can occur in a thermophilic organism. We have added some more text to address the issues of the

referee on page 10. However, given the space constraints we could not add an extended “Discussion”. As currently the RNAses / factors involved in degradation of RNA duplexes are unknown in *Sso*, a comparison of the eukaryotic miRNA mechanism with the situation in *Sso* would be highly speculative.

Which obvious questions need to be addressed next?

We briefly mentioned on page 10, last sentence of 2nd last paragraph that we aim at identifying RNAses involved in antisense regulation.

Referee #3

1. The significance of the work from a broader perspective is diminished by the finding (not clarified in the abstract) that the transcripts that purportedly regulate stability of the mRNA are very likely transposon derived. *Sso* is very atypical in that it has an extraordinary number of active transposable elements that result in large-scale genomic changes. Thus, the phenomenon under study may be a rare example of post-transcriptional control by non-coding RNAs as the result of recent events rather than a more deep rooted and pervasive gene expression regulation program commonly employed by archaea. For example, in related *Sulfolobus* species that lack large numbers of transposons is there evidence of a similar and conserved post-transcriptional regulation mechanism?

Our intention was to show that RNA based regulation occurs in a thermophilic archaeon, which has so far not been addressed, rather than to establish “*a more deep rooted and pervasive gene expression regulation program commonly employed by Archaea*”.

In the revised version we mention on page 5, discuss on page 10, and show in Fig S1D that the targeting sequence of RNA-257_{1,4}, i.e. the 3’ end and the 3’UTR of 1183 mRNA is only present in *Sso* but not in three other *Sulfolobales*. No ncRNAs have so far been identified in other *Sulfolobus* species. Our current knowledge on (putative) ncRNAs is therefore restricted to *S. solfataricus*. At present, studies in a related *Sulfolobus* species would have to start from scratch and can hardly be addressed in the timeframe given for the revision. Nevertheless, we still think that the story in *Sso* is interesting on its own, as it also provides an example of how regulatory ncRNAs can evolve. We discussed this on page ten and changed the title as to the origin of the antisense RNAs.

2. As the authors describe, the effect observed on ORF 1183 mRNA appears to result from convergent transcripts from immediately adjacent ORF 1182. The data presented does not exclude the possibility that the observed regulation is brought about via *cis* rather than (claimed) *trans* regulation. The observation that the effect can be artificially recapitulated in *trans* (Fig 3) with a reporter system does not speak to if the effect acts in *trans* physiologically. As the authors concede, a more definitive test (systematic gene knock-out of each species of RNA-2571-4) is currently not feasible.

We have stated on page 6 that we cannot exclude regulation in *cis* of 1183 mRNA. This was the reason for transplanting the 1183-3’UTR to *lacS* as well as to ORF 6454 (see new Fig S3). We have not claimed that 1183 mRNA is regulated in *trans*, we rather provide evidence that regulation by antisense RNA, as exemplified with the “reporter constructs” is feasible in *trans*.

3. No work was reported to address if the effects on gene expression observed under conditions of phosphate starvation or abundance occurs in vivo by transcriptional (rather than post-transcriptional) mechanisms differentially operating on the two convergent promoters that separately regulate expression of RNA-2571 or ORF1183 mRNA. Moreover, the *lacS* reporter assay should be performed under conditions of both phosphate abundance and starvation to address if the system recapitulates the regulation observed in vivo (Fig 2). Moreover, Northern probes that detect all of the 4 size forms would address if each RNA responds to phosphate limitation as does the *cis*-encoded RNA-2571 form that was specifically examined.

We have stated in the text that for 1183 mRNA we cannot exclude regulation at the transcriptional level (see page 6). Therefore, we used the “*lacS* reporters” that are under control of the same promoter (arabinose inducible promoter) and induced RNA257₁₋₄ depletion by phosphate limitation. *Sso* cells expressing the *lacS*-3'UTR-1183 and *lacS*-3'UTR-1183R26 transcripts were grown in full medium (+P) and under phosphate limiting conditions (-P) and the *lacS* mRNA levels were determined using RT-PCR (Fig. 3D). For both “*lacS*-3'UTR-1183 reporters” we observed elevated levels under phosphate limiting conditions.

We have also used RNA-257₃ and RNA-257_{2/4} specific probes and showed that the steady state levels of RNA-257₂, RNA-257₃ and RNA-257₄ are likewise decreased under phosphate limiting conditions (Fig. S1C).

4. In Fig 4, the stability of 1183 mRNA was found to be higher if it was not hybridized to RNA-2571 duplex when exposed to cell-free extract indicating a possible mechanism of action through degradation by an unknown double-strand specific RNA endonuclease. The analysis would have been more informative if the decay of end-labeled or (best) body labeled full-length mRNA was carried out rather than the indirect method (Northern analysis probing specific region of the mRNA). The expected mRNA decay intermediates that would have been possible to observe using this approach and a characterization of these breakdown products could be informative with regard to differential stability observed between free and duplexed mRNA.

We have performed an additional “*in vitro* degradation experiment” (see Fig. S5) with a body-labeled 3' fragment of the *lacS*-3'UTR-1183 transcript. While the full size *lacS*-3'UTR-1183 transcript was still present in *Sso* extracts after 25 min the *lacS*-3'UTR-1183:RNA-257₁ duplex was almost completely degraded after 20 min (see Fig S5). These experiments support those shown in Fig. 4 in that the “*lacS* reporter” is faster degraded when in duplex with RNA-257₁. Under the *in vitro* conditions, we could not discern specific decay intermediates.

2nd Editorial Decision

25 January 2013

Thank you for the submission of your revised manuscript to our journal. Since my colleague Alejandra Clark is currently not in the office, I have taken over the handling of your manuscript. We have now received the enclosed reports from the referees that were asked to re-assess it.

As you will see, referee 2 still has a few suggestions for how the manuscript could be further improved. I think that all her/his concerns regarding both the manuscript text/organization and further experimentation/quantification should be addressed before we can proceed with the official acceptance of your paper. We will therefore give you the exceptional opportunity for a second round of revision. Please also include the definition of the error bars for figure 4C.

Please note that it is EMBO reports policy that manuscripts need to be accepted at the latest 6 months after a first decision was made. In your case, this means that the manuscript needs to be accepted before the 11th of April. I therefore suggest that you submit the revised version that addresses the referee concerns - along with a point-by-point response - latest at the beginning of March, in order to give referee 2 some time to look at the final version.

Please let me know if you have any further comments or questions.

I look forward to seeing a revised form of your manuscript when it is ready.

REFeree REPORTS:

Referee #1:

The authors have satisfactorily addressed my comments.

I am still not convinced though that Figure 1 fully conveys the novelty of this paper, i.e. that the small RNAs act in trans. To a non-specialist reader, this will look like regulation by cis-encoded antisense RNAs that overlap the 3' end of ORF1183. Up to the authors!

Referee #2:

In the revised version of their manuscript now entitled "Antisense regulation by transposon derived RNAs in the hyperthermophilic archaeon *Sulfolobus solfataricus*" Maertens et al address most of the issues raised by the referees. They do additional, plasmid-based experiments to show that antisense regulation is feasible 'in trans' in a thermophile archaeon, *Sulfolobus solfataricus* (Sso). This is a significant finding of general interest. Furthermore it is plausible that the described regulation also takes place in the natural setting. Importantly, the authors clarify that they cannot exclude a cis-acting mechanism or a direct, phosphate-dependent regulation of Sso 1183 and they also point out that the presented example of an antisense regulation is transposon-derived and hence not necessarily general for all archaea.

Major points that still need to be addressed:

1. The finding that all four RNA257(1-4) transcripts are phosphate dependent is confusing. In a final discussion, the authors should comment on how they imagine this is achieved or how this regulation could have evolved four times independently.

2. The authors concede that for the natural setting they cannot exclude a cis-acting mechanism or a direct, phosphate-dependent regulation of Sso 1183. I would suggest, however, to move these statements to the final discussion on p.10. At the present place in the manuscript (p 6. last paragraph) it is not clear for the reader that these are final statements; rather the reader expects that the following experiments (p. 7) would address the conceded limitations. This is confusing.

3. The authors are strongly encouraged to include a final paragraph labelled 'Discussion' or 'Conclusions', where they reflect on the limitations, significance and generality of their findings. This is necessary to help the reader appreciate the results. These 'Conclusion' would comprise the present last two paragraphs (p9./10, starting with the question of possible RNases), but also the statements of the authors that they cannot exclude a cis-acting mechanism or a direct, phosphate-dependent regulation of Sso 1183 in the natural setting (p 6. last paragraph). Furthermore, there should be a comment on whether and how a phosphate-dependent regulation of RNA-257(1-4) would have evolved four times independently and in four different loci?

4. As previously suggested, the quantifications of RNA transcripts in Figures 3C,D, S2C, S3C would be more convincing if they were done by Northern blotting or qPCR (see Figure 2), rather than by simple RT-PCR. It is rather difficult to obtain a linear response in simple RT PCR.

5. Figure S2 A/B: Important: The authors need to show b-galactosidase activity for pMJ05-1183 under limiting phosphate conditions; i.e. that protein expression from pMJ05-1183 is phosphate-dependent. Ideally one would quantify the RNA transcripts (qPCR, Northern) and protein expression of ALL constructs under BOTH phosphate conditions and present it in a SINGLE Figure (somewhat similar to Figure S3). Figure S2C-legend: 'Synthesis' is probably not the correct term.

6. Figure S3: This is largely redundant with FigureS2 once the experiments for FigureS2 are completed. S3B: The Western blots should be quantified as well. S3C: The protein levels should be included in the graph. All values for P+ normalized to 100, values for P- in response.

7. Figure S4: It is not clear what can be concluded from this experiment. Naturally, if two complementary RNA sequences of > 50 nt in length are mixed in vitro, they will form a very stable duplex. To address the stability of such a duplex it is common to do a melting curve, e.g. in a qPCR machine. Gel analysis seems inappropriate, because samples cool down and reanneal during transfer into the slots of the gel, or in the slots while the gel is being loaded if buffers and gel itself are not kept at 75 C?

8. Figure S5: This experiment is of poor technical quality and does not add additional insight in its present form. I would suggest to remove it. (The gel is supposed to be denaturing and hence should not show any duplex species. Size markers are missing.) One would expect the accumulation of the single-stranded, central part of the lacS fusion with time; at least a smear or a shadow of it. Alternatively, one could have spotted some single-stranded RNA into the reaction, corresponding only to the central part of the lacS fusion to demonstrate that this is indeed degraded much more slowly.

Below are specific comments on the authors' response:

Editor:

1. All referees stress that further experiments are required to demonstrate that RNA-257-1 is a phosphate-dependent regulatory RNA. Specifically, they suggest that the lacS reporter assay should be performed under conditions of phosphate abundance and starvation.

Experiments addressing these questions are now shown in Fig 3D.

> O.K., Figure 3D shows phosphate dependent variation of RNA-2571 and pMJ05-1183 transcripts. The variations are negatively correlated and, for lacS transcripts, depend on the presence of the 1183 3'UTR. This shows that antisense regulation is feasible 'in trans' in Sso.

> As previously suggested, the quantifications of RNA transcripts in Figures 3C,D, S2C, S3C would be more convincing if they were done by Northern blotting or qPCR (see Figure 2), rather than by simple RT-PCR. It is rather difficult to obtain a linear response in simple RT PCR.

> The finding that antisense regulation is feasible 'in trans' in a thermophile archaeon is a significant finding of general interest.

2. In addition, since RNA-257-1 is derived from transposons, and Sso is atypical in that it contains a large number of transposable elements, all reviewers indicate that the generality of the described mechanism of post-transcriptional regulation by a trans-acting anti-sense RNA should be at least discussed, or better, if possible, experimentally addressed. For instance, referee #3 suggests that a related *Sulfolobus* species that lack such large number of transposons should be investigated for the presence of a post-transcriptional regulation mechanism.

We mention on page 5, discuss on page 10, and show in Fig S1D that the targeting sequence of RNA-2571-4, i.e. the 3' end and the 3'UTR of 1183 mRNA is only present in Sso but not in three other *Sulfolobales*. No ncRNAs have so far been identified in other *Sulfolobus* species. Our current knowledge on (putative) ncRNAs is therefore restricted to *S. solfataricus*. At present, studies in a related *Sulfolobus* species would have to start from scratch and can hardly be addressed in the timeframe given for the revision. Nevertheless, we still think that the story in Sso is interesting on its own, as it also provides an example of how regulatory ncRNAs can possibly evolve. We discussed this on page ten and changed the title as to the origin of the antisense RNAs.

> I tend to agree with the authors; the changes to the manuscript in order to point out the specialties of Sso are adequate.

3. Importantly, referees #2 and #3 are concerned that the current evidence based on a lacS reporter assay does not directly demonstrate anti-sense regulation by the trans-acting RNA *in vivo*, and suggest that further experiments to distinguish between cis- and trans-acting mechanisms should be performed.

To verify the findings with the lacS-3'UTR-1183 construct, we have performed an additional study with a similar 6454-3'UTR-1183 construct. ORF 6454 encodes the Sso Sm1 protein. In this case we were able to demonstrate a correlation between the intracellular levels of Sm1 protein, 6454 mRNA and RNA-2571 (please see ms, page 8 and Fig S3). As for the lacS-3'UTR-1183 RNA, a reasonable assumption is that RNA-2571 acts *in trans* to bring about the observed regulatory effect.

> The finding that antisense regulation is feasible 'in trans' in a thermophile archaeon is a significant finding of general interest. Furthermore, it is plausible that the described regulation also takes place in the natural setting. The experiments with the Sm1 protein, however, do not distinguish between cis- and trans-acting mechanisms but rather only confirm the results obtained with the lacS-fusions.

> Regarding the natural setting, the authors concede that they cannot exclude a cis-acting mechanism or a direct, phosphate-dependent regulation of Sso 1183 (p 6. last paragraph). I would suggest, however, to move these statements to the final discussion on p.10.

4. Finally, all reviewers point to additional issues including technical concerns and further controls. Please see at the respective comments of the reviewers.

Referee#1:

1. The presented evidence that RNA-257/1 binds the 3' UTR of the 1183 mRNA is cumulative, i.e. represents the sum of indirect results which include *in silico* prediction of a long antisense interaction, disruption of this putative interaction *in vivo*, and complex formation (gel shifts experiments) *in vitro*. The lack of a good experimental system that would not allow the authors to introduce compensatory base pair changes to validate base pairing *in vivo*, which is the gold standard in eubacteria. However, the authors present in Figure 4 a lysate-based RNA stability assay that would lend itself as a surrogate system to study the effects of mutations in the predicted RNA duplex and seek restoration of base pairing by compensatory mutations in the sRNA and the target.

We have introduced deletions (25 bases) in 1183 mRNA as well as in RNA-2571. However, due to the long complementarity between both RNAs, we still observed duplex formation in band shift assays when we incubated wt 1183 RNA with mutated RNA-2571 or vice versa when mutated 1183 RNA was incubated with wt RNA -2571. Presumably as a result, we did not observe significant differences in the stability of the 3' end of 1183 mRNA when the respective 1183:RNA-2571 duplexes were incubated in Sso extracts. Another complication towards observing differences in "stability" with the mutant RNAs is the short half life of the 3' end of 1183 mRNA (see Fig. 4C) when in duplex with RNA-2571.

This issue is also obvious from our studies with plasmid pMJ05-1183-R26 (see Fig 3C). Although 26 nt were replaced in the construct to diminish binding of RNA-2571-4, we still observe reduced levels of lacS-3'UTR-1183R26 RNA, when compared with lacS (with its authentic 3'UTR) or lacS Δ (Fig. 3C) indicating that RNA-2571-4 can still base-pair with lacS-3'UTR-1183R26.

2. The *in vitro* RNA stability results presented in Figure 4 are difficult to interpret with respect to reproducibility. The experiments should be performed in triplicates, and RNA stability should be presented as graphs to be able to calculate RNA half-lives and judge the variation in the assay. In addition, as suggested above, informative mutant versions of the interacting RNAs should be included.

We performed the experiment with the 1183 mRNA +/- RNA-257 in triplicate and quantified the blots with image quant. The resulting graph is now shown in Fig 4C. To verify these data we also performed similar assays with body-labeled lacS'-3'UTR-1183 RNA (see Fig S5).

3. Figure 3: Is the downregulation of the 3'UTR reporters phosphate-dependent? In other words, would phosphate-starvation repressing the small RNAs elevate the expression of the pJM105-1183 and pJM105-1183-R26 reporters? This would be important to convince the reader that these reporters fully recapitulate the small RNA-mediated repression of the native 1183 mRNA.

As suggested, we repeated the experiment and grew the *Sso* cells expressing the lacS-3'UTR-1183 and lacS-3'UTR-1183R26 transcripts in full medium (+P) and under phosphate limiting conditions (-P). Using RT-PCR the lacS mRNA levels were determined (Fig. 3D). For both "lacS- constructs" we observed elevated levels during phosphate limiting conditions, i.e. under conditions when the levels of RNA-2571-(4) are decreased (see Fig. 3D and Fig S1C)

Minor issues:

4. Page 3, Introduction: I recommend this part of the manuscript be used to make clear that targeting the 5' UTR versus 3' UTR has been a major difference between prokaryotes and eukaryotes, for the general reader to appreciate the key novelty of the present work.

To make this more explicit one sentence was added at the end of the first paragraph on page 3.

5. Page 5, middle part: Does the listing of other families of transposon-derived small RNA genes really contribute to an understanding of the ncRNA-257 family in the context of the current paper?

As suggested and given the space constraints, we have eliminated the information on RNA-75 and RNA-108.

6. Page 6, line 15, "... were increased and decreased in the presence and absence of phosphate": Since the authors examine only two conditions, it is hard to say whether there was an increase AND a decrease. If phosphate-replete is the reference condition, it will do to say that a decrease was observed in the absence of phosphate.

We have changed the language according to the suggestion of the referee by stating were decreased under phosphate limiting conditions...

7. Page 8, last paragraph of main text: After presenting the results for the RNA stability, the text jumps to a discussion of nucleases that might be involved in the regulation. It would be helpful, however, to briefly summarize the results of this work, and critically reflect the evidence for and potential generality of the reported regulation in the 3' UTR.

We have done so and have also added more text on page 10 with regard to the possible origin of ncRNAs.

8. Figure 1 requires improvement. It is difficult to understand where the RNA-257 copies are located (are they really 300kb long?) and whether or not RNA-257-1 is genetically linked with the green Tn. Figure 1 was modified.

Referee #2:

1. It is unclear to which degree the described effects are phosphate-specific. The authors should try and demonstrate this more thoroughly.

Please see our response to comment 3 of referee #1 and below.

1.1 They should make available the results of the tested growth/ stress conditions as part of the Supplement to illustrate that phosphate levels are indeed the primary cause of the different RNA-257-1 levels. Add phosphate concentrations to this Supplemental Figure (and to Figure 2).

We tested the abundance of RNA-2571 in logarithmic, in stationary phase and during nutrient starvation. Under these conditions we did not observe any significant differences in full medium (+P). As these blots only show an equal abundance and the number of suppl. Figs. is limited to 5, we preferred to rather show the phosphate dependent abundance of RNA-2572-4 in Fig S1C.

The KH₂PO₄ concentration in full medium was 280 mg/l. The cells were then diluted in medium

without KH₂PO₄. We have added more information on "phosphate limiting conditions in Suppl. Information and added the concentration of KH₂PO₄ in the legend to Fig 2.

> O.K., but see comments on 1.2

1.2 Include titration experiments at several intermediate phosphate concentrations and analyze RNA levels by qPCR as in Figure 2B. Show phosphate response curves for RNA-257-1, Sso1183, Sso0412 and include the Sso3019 (lacS) constructs pMJ05-1183-R26 and pMJ05 (and/or pMJ05-delta). If possible, also include RNA-257-2, RNA-257-3, and RNA-257-4. It would be surprising if the transcription of all four versions of RNA-257 were phosphate-dependent. The authors should explain to the reader why they focused exclusively on RNA257-1. Furthermore, it is crucial for the proposed regulation 'in trans' to demonstrate that the amount of Sso3019 RNA as derived from the pMJ05-1183-R26 plasmid (Figure 3 and Point 2.2) is phosphate-dependent.

We have only determined the levels of RNA-2571-4 in full Brock's medium and Brocks medium depleted for phosphate. We used RNA-2573 and RNA-2572/4 specific probes to test if the paralogues RNAs are also down-regulated under phosphate limiting conditions. As shown in Fig S1C, the steady state levels of RNA-2572, RNA-2573 and RNA-2574 are likewise decreased under these conditions.

> It would be more informative to quantify the upregulation of RNA-2571-4. Furthermore, I find it confusing and I wonder whether the authors believe that all four isoforms have independently evolved a phosphate-dependent transcription? Would such redundancy make sense in the proposed regulatory pathway? The authors should comment on these points in the final discussion on p.10.

As suggested we show in Fig. 3D that the amount of of Sso 3019 mRNA is phosphate dependent when expressed in strains PH1-16(pMJ05-1183) and PH1-16 (pMJ05-1183R26), respectively. While the level of Sso 3019 mRNA was increased during phosphate limitation, the levels of RNA-2571 were decreased (Fig 3D). In contrast, the abundance of the two control mRNAs 0412 and 0764 was the same regardless whether the cells were grown in full medium or in medium depleted for phosphate (Fig 3D). We focused on RNA-2571 being one representative of RNA-2571-4.

> O.K.

1.3 If feasible, add phosphate to the medium and follow RNA levels by qPCR in a time-course experiment. This might demonstrate that levels of RNA257-1 rise BEFORE levels of Sso1183 drop and it might help to establish cause and effect. In any case, the authors are encouraged to explain to the reader why they do not consider the alternative hypothesis that Sso1183 levels are regulated independently of RNA-257-1 and simply have an effect on RNA-257-1 levels as a consequence of the base-pairing (e.g. use the amounts of RNA in Figure 2B as an argument).

In the revised version we have made these concerns apparent for the reader on page 6. To verify the findings obtained with 1183, we transplanted the 1183 3'UTR to the 3'end of lacS, expression of which (lacS only) is independent of phosphate (see Fig 2B). Only when the 1183 3'UTR was fused to lacS, the abundance of the lacS-3'UTR-1183 became phosphate dependent (Fig 3D).

> O.K., the authors concede that for the natural setting they cannot exclude a cis-acting mechanism or a direct, phosphate-dependent regulation of Sso 1183 (p 6. last paragraph). I would suggest, however, to move these statements to the final discussion on p.10, where the authors are encouraged to reflect on the limitations, significance and generality of their findings. At the present place in the manuscript (p 6. last paragraph) it is not clear for the reader that these are final statements; rather the reader expects that the following experiments (p. 7) would address the conceded limitations.

Moreover, to verify the findings with the lacS-3'UTR-1183 construct, we have performed an additional study with a similar 6454-3'UTR-1183 construct. ORF 6454 encodes the Sso Sm1 protein. In this case we were able to demonstrate a correlation between the intracellular levels of Sm1 protein, 6454 mRNA and RNA-2571 (please see ms., page 8 and Fig S3).

> O.K., but the experiments are simply redundant with the lacS -fusions?

2. The experiment in Figure 3 (transfer of the Sso1183 3'UTR to a lacS reporter) is an important and crucial experiment to demonstrate that RNA257-1 can work in trans. The experiment should include additional controls.

2.1 It may be problematic to introduce transposon-derived 1182 DNA on the pMJ05 plasmid. This could potentially trigger some host transposon defense mechanisms or recombination events that prevent the lacS fusion RNA to be properly transcribed in the first place. This would provide an

alternative explanation for the complete absence of evidence on the protein and RNA levels for pMJ05-1183. Is there a way to check for plasmid integrity or to control for such effects?

We introduced the 3' UTR of 1183 that is partially complementary to the transposon ORF 1182. We confirmed the presence / integration of the plasmid after the transformation of *Sso* cells growing from single colonies, which we used for this study. To make this clear we have added more text to Suppl. Information (see: Transformation of *Sso*).

> O.K.

We could confirm the transcription, as low RNA levels of lacS mRNA became detectable if we used more cDNA (500 ng) for the PCR; the experiment is now shown in Fig. S2C.

> O.K.

2.2 The R26 mutation of pMJ05-1183 apparently attenuates regulation by RNA-257-1, such that pMJ05-1183-R26 RNA gets detectable by RT PCR (Figure 3C), but the low amount of pMJ05-1183-R26 RNA indicates that is still under control of RNA-257-1 due to partial complementarity. If this is the case, levels of pMJ05-1183-R26 RNA should be phosphate dependent (see the crucial experiment suggested under Point 1.2). The authors should demonstrate phosphate dependence of pMJ05-1183-R26 RNA to support their claim that phosphate-dependent regulation of *Sso*1183 RNA occurs via RNA-257-1.

We have performed the suggested experiment (see Fig 3D). *Sso* cells expressing the lacS-3'UTR-1183 and lacS-3'UTR-1183R26 transcripts were grown in full media (+P) and under phosphate limiting conditions (-P) and the lacS mRNA levels were determined using RT-PCR. For both constructs we see elevated levels of lacS during phosphate limitation. See text, page 7 and 8.

> O.K.

3. The rationale and motivation for the *in vitro* experiments in Figure 4 are not very clear. At best, one can conclude that *S. solfataricus* contains double strand-specific RNases, but if the presented experiments recapitulate the physiological mechanism of *Sso*1183 RNA degradation in any way remains doubtful. It may be more helpful to move Figure 4 to the Supplement and rather investigate first how pMJ05-1183-R26 RNA (Figure 3C) is degraded *in vivo*. To this aim one could do Northern blots to see how much of pMJ05-1183-R26 RNA is full-length and whether there are specific degradation intermediates (e.g. whether the complementary part of 3'UTR is degraded preferentially). Alternatively, this could be done by qPCR, targeting different regions of the mRNA. Fig 4: Although we observed a correlation between the levels of RNA-257 and the "reporter RNAs" we sought a means to show that binding of RNA-257 to the complementary stretch present at the 3' end and in the 3' UTR results in faster degradation of this part.

> O.K., the double-stranded part is degraded faster than the single-stranded part. Please include lanes 3 and 4 in Figure 4C for comparison.

We have performed an additional "in vitro degradation experiment" (see Fig. S5) with a body-labeled 3' fragment of the lacS'-3'UTR-1183 transcript (see referee #3, point 4; Fig S5). This experiment revealed that the part of lacS'-3'UTR-1183 complementary to RNA2571 is faster degraded *in vitro* than the lacS'-3'UTR-1183 in the absence of RNA-2571 (see Fig S5). However, we could not discern specific degradation intermediates under these conditions.

> This experiment is of poor technical quality and does not add additional insight in its present form. I would suggest to remove it. (The gel is supposed to be denaturing and hence should not show any duplex species. Size markers are missing.) One would expect the accumulation of the single-stranded, central part of the lacS fusion with time; at least a smear or a shadow of it. Alternatively, one could have spotted in some single-stranded RNA corresponding only to the central part of the lacS fusion to demonstrate that this is indeed degraded much more slowly.

4. To stimulate the interest of a more generally interested audience, the authors should provide a more comprehensive discussion of how unique/ significant/ widespread the described regulatory mechanism might be in archaea and what are the obvious DIFFERENCES to the eukaryotic microRNA mechanisms. Which obvious questions need to be addressed next? How do the authors see the transposon-derived origin of RNA-257-1? Is it a problem for the generalization of the mechanism?

Here, we only aimed at providing evidence that antisense regulation can occur in a thermophilic

organism. We have added some more text to address the issues of the referee on page 10. However, given the space constraints we could not add an extended "Discussion". As currently the RNases / factors involved in degradation of RNA duplexes are unknown in *Sso*, a comparison of the eukaryotic miRNA mechanism with the situation in *Sso* would be highly speculative.

> The authors are strongly encouraged to include a final paragraph labelled 'Discussion' or 'Conclusions', where they reflect on the limitations, significance and generality of their findings. This is necessary to help the reader appreciate the results. These 'Conclusion' would comprise the present last two paragraphs (p9./10, starting with the question of possible RNases), but also the statements of the authors that they cannot exclude a cis-acting mechanism or a direct, phosphate-dependent regulation of *Sso* 1183 in the natural setting (p 6. last paragraph). Furthermore, there should be a comment on whether and how a phosphate-dependent regulation of RNA-257(1-4) would have evolved four times independently and in four different loci?

Which obvious questions need to be addressed next?

We briefly mentioned on page 10, last sentence of 2nd last paragraph that we aim at identifying RNases involved in antisense regulation.

Referee #3

1. The significance of the work from a broader perspective is diminished by the finding (not clarified in the abstract) that the transcripts that purportedly regulate stability of the mRNA are very likely transposon derived. *Sso* is very atypical in that it has an extraordinary number of active transposable elements that result in large-scale genomic changes. Thus, the phenomenon under study may be a rare example of post-transcriptional control by non-coding RNAs as the result of recent events rather than a more deep rooted and pervasive gene expression regulation program commonly employed by archaea. For example, in related *Sulfolobus* species that lack large numbers of transposons is there evidence of a similar and conserved post-transcriptional regulation mechanism?

Our intention was to show that RNA based regulation occurs in a thermophilic archaeon, which has so far not been addressed, rather than to establish "a more deep rooted and pervasive gene expression regulation program commonly employed by Archaea".

In the revised version we mention on page 5, discuss on page 10, and show in Fig S1D that the targeting sequence of RNA-2571-4, i.e. the 3' end and the 3'UTR of 1183 mRNA is only present in *Sso* but not in three other *Sulfolobales*. No ncRNAs have so far been identified in other *Sulfolobus* species. Our current knowledge on (putative) ncRNAs is therefore restricted to *S. solfataricus*. At present, studies in a related *Sulfolobus* species would have to start from scratch and can hardly be addressed in the timeframe given for the revision. Nevertheless, we still think that the story in *Sso* is interesting on its own, as it also provides an example of how regulatory ncRNAs can evolve. We discussed this on page ten and changed the title as to the origin of the antisense RNAs.

> O.K., changes in the discussion and title were made to address these concerns. The findings are limited to *Sso* so far.

2. As the authors describe, the effect observed on ORF 1183 mRNA appears to result from convergent transcripts from immediately adjacent ORF 1182. The data presented does not exclude the possibility that the observed regulation is brought about via cis rather than (claimed) trans regulation. The observation that the effect can be artificially recapitulated in trans (Fig 3) with a reporter system does not speak to if the effect acts in trans physiologically. As the authors concede, a more definitive test (systematic gene knock-out of each species of RNA-2571-4) is currently not feasible.

We have stated on page 6 that we cannot exclude regulation in cis of 1183 mRNA. This was the reason for transplanting the 1183-3'UTR to lacS as well as to ORF 6454 (see new Fig S3). We have not claimed that 1183 mRNA is regulated in trans, we rather provide evidence that regulation by antisense RNA, as exemplified with the "reporter constructs" is feasible in trans.

> The finding that antisense regulation is feasible 'in trans' in a thermophile archaeon is a significant finding of general interest. Furthermore, it is plausible that the described regulation also takes place in the natural setting, but the authors concede that there may be alternative mechanisms in the natural setting. These statements should be moved to the final discussion, however.

3. No work was reported to address if the effects on gene expression observed under conditions of

phosphate starvation or abundance occurs *in vivo* by transcriptional (rather than post-transcriptional) mechanisms differentially operating on the two convergent promoters that separately regulate expression of RNA-2571 or ORF1183 mRNA. Moreover, the lacS reporter assay should be performed under conditions of both phosphate abundance and starvation to address if the system recapitulates the regulation observed *in vivo* (Fig 2). Moreover, Northern probes that detect all of the 4 size forms would address if each RNA responds to phosphate limitation as does the cis-encoded RNA-2571 form that was specifically examined.

We have stated in the text that for 1183 mRNA we cannot exclude regulation at the transcriptional level (see page 6). Therefore, we used the "lacS reporters" that are under control of the same promoter (arabinose inducible promoter) and induced RNA2571-4 depletion by phosphate limitation. Sso cells expressing the lacS-3'UTR-1183 and lacS-3'UTR-1183R26 transcripts were grown in full medium (+P) and under phosphate limiting conditions (-P) and the lacS mRNA levels were determined using RT-PCR (Fig. 3D). For both "lacS-3'UTR-1183 reporters" we observed elevated levels under phosphate limiting conditions.

> O.K, the authors concede that for the natural setting they cannot exclude a cis-acting mechanism or a direct, phosphate-dependent regulation of Sso 1183 (p 6. last paragraph). I would suggest, however, to move these statements to the final discussion on p.10, where the authors are encouraged to reflect on the limitations, significance and generality of their findings. At the present place in the manuscript (p 6. last paragraph) it is not clear for the reader that these are final statements; rather the reader expects that the following experiments (p. 7) would address the conceded limitations.

> As previously suggested, the quantifications of RNA transcripts in Figures 3C,D, S2C, S3C would be more convincing if they were done by Northern blotting or qPCR (see Figure 2), rather than by simple RT-PCR. It is rather difficult to obtain a linear response in simple RT PCR.

We have also used RNA-2573 and RNA-2572/4 specific probes and showed that the steady state levels of RNA-2572, RNA-2573 and RNA-2574 are likewise decreased under phosphate limiting conditions (Fig. S1C).

> It would be more informative to quantify the upregulation of RNA-2571-4. Furthermore, I find it confusing and I wonder whether the authors believe that all four isoforms have independently evolved a phosphate-dependent transcription? Would such redundancy make sense in the proposed regulatory pathway? The authors should comment on these points in the final discussion on p.10.

4. In Fig 4, the stability of 1183 mRNA was found to be higher if it was not hybridized to RNA-2571 duplex when exposed to cell-free extract indicating a possible mechanism of action through degradation by an unknown double-strand specific RNA endonuclease. The analysis would have been more informative if the decay of end-labeled or (best) body labeled full-length mRNA was carried out rather than the indirect method (Northern analysis probing specific region of the mRNA). The expected mRNA decay intermediates that would have been possible to observe using this approach and a characterization of these breakdown products could be informative with regard to differential stability observed between free and duplexed mRNA.

We have performed an additional "in vitro degradation experiment" (see Fig. S5) with a body-labeled 3' fragment of the lacS-3'UTR-1183 transcript. While the full size lacS-3'UTR-1183 transcript was still present in Sso extracts after 25 min the lacS-3'UTR-1183:RNA-2571 duplex was almost completely degraded after 20 min (see Fig S5). These experiments support those shown in Fig. 4 in that the "lacS reporter" is faster degraded when in duplex with RNA-2571. Under the *in vitro* conditions, we could not discern specific decay intermediates.

> This experiment is of poor technical quality and does not add additional insight in its present form. I would suggest to remove it. (The gel is supposed to be denaturing and hence should not show any duplex species. Size markers are missing.) One would expect the accumulation of the single-stranded, central part of the lacS fusion with time; at least a smear or a shadow of it. Alternatively, one could have spotted in some single-stranded RNA corresponding only to the central part of the lacS fusion to demonstrate that this is indeed degraded much more slowly.

Referee#1:

I am still not convinced though that Figure 1 fully conveys the novelty of this paper, i.e. that the small RNAs act in *trans*. To a non-specialist reader, this will look like regulation by cis-encoded antisense RNAs that overlap the 3' end of ORF1183.

We have altered Fig 1 to show that the small RNAs act in *trans* on the *lacS-1183* transcript. In addition, we emphasized in 'Conclusion and Perspective' that RNA257₁₋₄ act in *trans* on the *lacS-1183/6454-1183* transcripts.

Referee#2:

1. The finding that all four RNA257(1-4) transcripts are phosphate dependent is confusing. In a final discussion, the authors should comment on how they imagine this is achieved or how this regulation could have evolved four times independently.

We have commented on this issue in the 'Results and Discussion' section, page 6, 1st paragraph. For phosphate regulated genes of *Mycobacterium smegmatis* [reference 17) inverted repeat regions have been described to be important. Inverted repeats are present in all four promoter regions (directly upstream of box A) of the RNA257₁₋₄ genes, which are now highlighted in Fig. S1B, and which could be remnants from transposition events. We have speculated on this in the text. Clearly, as the regulatory elements for phosphate dependent genes are unknown in *Sso*, it remains to be shown whether these elements are indeed important for phosphate dependent transcription.

2. The authors concede that for the natural setting they cannot exclude a cis-acting mechanism or a direct, phosphate-dependent regulation of *Sso* 1183. I would suggest, however, to move these statements to the final discussion on p.10. At the present place in the manuscript (p 6. last paragraph) it is not clear for the reader that these are final statements; rather the reader -+expects that the following experiments (p. 7) would address the conceded limitations. This is confusing.

We agree that the original sentence was confusing. We have modified the manuscript and have- by first stating that we cannot exclude a cis-acting mechanism (page 6, 2nd paragraph)- provided a rationale for continuing with the *lacS* constructs.

Later on (page 7, 3rd paragraph), we provide a rationale for looking at phosphate dependent regulation.

3. The authors are strongly encouraged to include a final paragraph labelled 'Discussion' or 'Conclusions', where they reflect on the limitations, significance and generality of their findings. This is necessary to help the reader appreciate the results. These 'Conclusion' would comprise the present last two paragraphs (p9./10, starting with the question of possible RNases), but also the statements of the authors that they cannot exclude a cis-acting mechanism or a direct, phosphate-dependent regulation of *Sso* 1183 in the natural setting (p 6. last paragraph). Furthermore, there should be a comment on whether and how a phosphate-dependent regulation of RNA-257(1-4) would have evolved four times independently and in four different loci?

As suggested, we added a paragraph 'Concluding remarks' and re-mentioned the shortcomings on " the cis-acting mechanism..." and ..."phosphate dependent regulation" of *Sso*1183..." at the beginning of this section. However, we found it appropriate to discuss the phosphate dependent regulation of RNA257₁₋₄ in the 'Results and Discussion' section (see point 1) after showing that the abundance of these RNAs is phosphate dependent.

4. As previously suggested, the quantifications of RNA transcripts in Figures 3C,D, S2C, S3C

would be more convincing if they were done by Northern blotting or qPCR (see Figure 2), rather than by simple RT-PCR. It is rather difficult to obtain a linear response in simple RT PCR.

As suggested we used qPCR to verify the data obtained by RT-PCR formerly shown in Fig 3C, D and S2C. The results are now presented in Fig. 3C, which replaces the former Fig 3D. The former Fig. 3C is now shown as Fig. S2B. As these new qPCR results reflect those formerly obtained by RT-PCR, and due to the time constraints, we did not apply qPCR to verify the results in former Fig S3C (now Fig 3B).

5. Figure S2 A/B: Important: The authors need to show b-galactosidase activity for pMJ05-1183 under limiting phosphate conditions; i.e. that protein expression from pMJ05-1183 is phosphate-dependent. Ideally one would quantify the RNA transcripts (qPCR, Northern) and protein expression of ALL constructs under BOTH phosphate conditions and present it in a SINGLE Figure (somewhat similar to Figure S3). Figure S2C-legend: 'Synthesis' is probably not the correct term.

As requested, we quantified the corresponding RNA transcripts under +P and - P conditions using qPCR. As can be seen from Fig. 3C, there is a phosphate dependent inverse correlation between the levels of *lacS-1183* and *lacS1183R26* and RNA257₁₋₄, as was also shown by RT-PCR in the last version of the ms. in Fig 3D. However, the β-galactosidase activities obtained with plasmids pMJ05-1183 and pMJ05-1183R26 (Fig S2A) were very low and not distinguishable when the cells were cultivated in the presence or in the absence of phosphate. Apparently there is enough RNA-257₁₋₄ under both conditions to drastically reduce the transcript levels of the "reporter transcripts" (see Fig 3D and Fig S2B), which results in rather low β-galactosidase activities (Fig S2A), and in turn precludes a distinctive read out dependent on the levels of RNA257₁₋₄. We have also tried to use antibodies to detect the LacS protein. However, their sensitivity was insufficient to detect minute amounts of the protein.

Nevertheless, a phosphate dependent inverse correlation between the levels of *6454-1183* and RNA257₁₋₄, is not only appreciable at the transcript levels (Fig S3B) but also at the level of the Sm1 protein (shown in Fig S3C). We mentioned this in the text.

6. Figure S3: This is largely redundant with FigureS2 once the experiments for FigureS2 are completed. S3B: The Western blots should be quantified as well. S3C: The protein levels should be included in the graph. All values for P+ normalized to 100, values for P- in response.

We agree that the results of the Sso *6454-1183* fusion (Fig S3) are somewhat redundant. However, they are supportive in terms of the *trans-acting* mechanism. In addition, as pointed out (point 6), phosphate dependent regulation of *6454-1183* by RNA257₁₋₄ was also reflected at the protein level. As this set of experiments supports the results obtained with the *lacS* fusions, we would like to retain Fig S3. The Fig was amended as suggested.

7. Figure S4: It is not clear what can be concluded from this experiment. Naturally, if two complementary RNA sequences of > 50 nt in length are mixed in vitro, they will form a very stable duplex. To address the stability of such a duplex it is common to do a melting curve, e.g. in a qPCR machine. Gel analysis seems inappropriate, because samples cool down and reanneal during transfer into the slots of the gel, or in the slots while the gel is being loaded if buffers and gel itself are not kept at 75 C?

As suggested, we performed a melting curve analysis to determine the stability of the duplex. The results are now presented as Fig S4B.

8. Figure S5: This experiment is of poor technical quality and does not add additional insight in its present form. I would suggest to remove it. (The gel is supposed to be denaturing and hence should

not show any duplex species. Size markers are missing.) One would expect the accumulation of the single-stranded, central part of the lacS fusion with time; at least a smear or a shadow of it. Alternatively, one could have spotted some single-stranded RNA into the reaction, corresponding only to the central part of the lacS fusion to demonstrate that this is indeed degraded much more slowly.

We removed the Fig as we also think that it is redundant with regard to Fig 4. In addition, we can not really explain why we detect duplex species in the denaturing gel.

3rd Editorial Decision

15 March 2013

Thank you for the submission of your revised manuscript. Referee 2 is overall happy with the manuscript now, however, s/he still has a few minor comments that should be addressed before we can proceed with the official acceptance of your manuscript.

I also would like to suggest some minor changes to the abstract, as follows:

We report the first example of antisense RNA regulation in a hyperthermophilic archaeon. In *Sulfolobus solfataricus* the transposon derived paralogous RNAs, RNA-2571-4, exhibit extended complementarity to the 3'UTR of the 1183 mRNA, encoding a putative phosphate transporter. Phosphate limitation results in decreased RNA-2571 and increased 1183 mRNA levels. Correspondingly, the 1183 mRNA is faster degraded *in vitro* upon duplex formation with RNA-2571. Insertion of the 1183 3'UTR downstream of the lacS gene results in strongly reduced lacS mRNA levels in transformed bacteria, indicating that antisense regulation can function *in trans*.

Please let me know if you agree with these changes.

I look forward to seeing a new revised version of your manuscript as soon as possible.

REFEREE REPORT:

Referee #2:

The clarity of the manuscript by Märtens et al has improved significantly; the additional experiments clarified remaining questions and add to the value of this work. I have two minor last comments that I trust the authors can address without the need for an additional review:

1. On p.9. the authors claim that the melting curve 'revealed that the duplex is stable within a temperature range from 75 deg to 90 deg'. I do not think this is entirely correct. In my impression, the curve in Fig S4B rather shows that the RNA structure is stable below 75 deg and begins to melt at temperatures above 75 deg, but that full denaturation requires temperatures above 90 deg. A control for the 1183-3 UTR RNA alone should be added to Fig S4B, if available.

2. In Fig S3A the bars could be arranged slightly differently and the legend expanded for an explanation of the green bar. It is not immediately clear that the green bar is protein, or that RNA and protein levels are addressed in this figure. I would suggest to order the bars red, blue, green (instead of blue, red, green) and use as labels: red (RNA-257-1), blue (Sso 6454-1183 -- RNA), green (Sso 6454-1183 -- Protein).

A typo p.7 'extent' not 'extend'

Thank you for your quick reply. In the final version we have made the minor changes as requested by you and those made by referee #2. Thank you very much for your support during the review process.

Your suggestion:

We have changed the *abstract* as specified, except that we used the term "...transformed cells" instead of "...transformed bacteria".

Referee#2:

Melting curve: We changed in the text the wording as specified by the reviewer, we added the curve for 1183-3'UTR alone as control.

Fig S3: The Fig. was modified as suggested.

I am very pleased to accept your manuscript for publication in the next available issue of EMBO reports. Thank you for your contribution to our journal.

As part of the EMBO publication's Transparent Editorial Process, EMBO reports publishes online a Review Process File to accompany accepted manuscripts. As you are aware, this File will be published in conjunction with your paper and will include the referee reports, your point-by-point response and all pertinent correspondence relating to the manuscript.

If you do NOT want this File to be published, please inform the editorial office within 2 days, if you have not done so already, otherwise the File will be published by default [contact: emboreports@embo.org]. If you do opt out, the Review Process File link will point to the following statement: "No Review Process File is available with this article, as the authors have chosen not to make the review process public in this case."

Finally, we provide a short summary of published papers on our website to emphasize the major findings in the paper and their implications/applications for the non-specialist reader. To help us prepare this short, non-specialist text, we would be grateful if you could provide a simple 1-2 sentence summary of your article in reply to this email.

Thank you again for your contribution to EMBO reports and congratulations on a successful publication. Please consider us again in the future for your most exciting work.